# ON GRAPH NEURAL NETWORKS VERSUS GRAPH-AUGMENTED MLPS

**Lei Chen**[*]**, Zhengdao Chen**[*]
Courant Institute of Mathematical Sciences
New York University, New York, NY
{lc3909, zc1216}@nyu.edu

**Joan Bruna**
Courant Institute of Mathematical Sciences
Center for Data Science
New York University, New York, NY
bruna@cims.nyu.edu

## ABSTRACT

From the perspectives of expressive power and learning, this work compares multi-layer Graph Neural Networks (GNNs) with a simplified alternative that we call Graph-Augmented Multi-Layer Perceptrons (GA-MLPs), which first augments node features with certain multi-hop operators on the graph and then applies learnable node-wise functions. From the perspective of graph isomorphism testing, we show both theoretically and numerically that GA-MLPs with suitable operators can distinguish almost all non-isomorphic graphs, just like the Weisfeiler-Lehman (WL) test and GNNs. However, by viewing them as node-level functions and examining the equivalence classes they induce on rooted graphs, we prove a separation in expressive power between GA-MLPs and GNNs that grows exponentially in depth. In particular, unlike GNNs, GA-MLPs are unable to count the number of attributed walks. We also demonstrate via community detection experiments that GA-MLPs can be limited by their choice of operator family, whereas GNNs have higher flexibility in learning.

## 1 INTRODUCTION

While multi-layer Graph Neural Networks (GNNs) have gained popularity for their applications in various fields, recently authors have started to investigate what their true advantages over baselines are, and whether they can be simplified. On one hand, GNNs based on neighborhood-aggregation allows the combination of information present at different nodes, and by increasing the depth of such GNNs, we increase the size of the receptive field. On the other hand, it has been pointed out that deep GNNs can suffer from issues including over-smoothing, exploding or vanishing gradients in training as well as bottleneck effects (Kipf & Welling, 2016; Li et al., 2018; Luan et al., 2019; Oono & Suzuki, 2020; Rossi et al., 2020; Alon & Yahav, 2020).

Recently, a series of models have attempted at relieving these issues of deep GNNs while retaining their benefit of combining information across nodes, using the approach of firstly augmenting the node features by propagating the original node features through powers of graph operators such as the (normalized) adjacency matrix, and secondly applying a node-wise function to the augmented node features, usually realized by a Multi-Layer Perceptron (MLP) (Wu et al., 2019; NT & Maehara, 2019; Chen et al., 2019a; Rossi et al., 2020). Because of the usage of graph operators for augmenting the node features, we will refer to such models as *Graph-Augmented MLPs (GA-MLPs)*. These models have achieved competitive performances on various tasks, and moreover enjoy better scalability since the augmented node features can be computed during preprocessing (Rossi et al., 2020). Thus, it becomes natural to ask what advantages GNNs have over GA-MLPs.

In this work, we ask whether GA-MLPs sacrifice expressive power compared to GNNs while gaining these advantages. A popular measure of the expressive power of GNNs is their ability to distinguish non-isomorphic graphs (Hamilton et al., 2017; Xu et al., 2019; Morris et al., 2019). In our work, besides studying the expressive power of GA-MLPs from the viewpoint of graph isomorphism tests, we propose a new perspective that better suits the setting of node-prediction tasks: we analyze the

---

[*]Equal contributions. Code available at https://github.com/leichen2018/GNN_vs_GAMLP.

expressive power of models including GNNs and GA-MLPs as node-level functions, or equivalently, as functions on rooted graphs. Under this perspective, we prove an exponential-in-depth gap between the expressive powers of GNNs and GA-MLPs. We illustrate this gap by finding a broad family of user-friendly functions that can be provably approximated by GNNs but not GA-MLPs, based on counting attributed walks on the graph. Moreover, via the task of community detection, we show a lack of flexibility of GA-MLPs, compared to GNNs, to learn the best operators to use.

In summary, our main contributions are:

- Finding graph pairs that several GA-MLPs cannot distinguish while GNNs can, but also proving there exist simple GA-MLPs that distinguish almost all non-isomorphic graphs.
- From the perspective of approximating node-level functions, proving an exponential gap between the expressive power of GNNs and GA-MLPs in terms of the equivalence classes on rooted graphs that they induce.
- Showing that the functions that count a particular type of attributed walk among nodes can be approximated by GNNs but not GA-MLPs both in theory and numerically.
- Through community detection tasks, demonstrating that GNNs have higher flexibility in learning than GA-MLPs due to the fixed choice of the operator family in the latter.

## 2 RELATED WORKS

**Depth in GNNs** Kipf & Welling (2016) observe that the performance of Graph Convolutional Networks (GCNs) degrade as the depth grows too large, and the best performance is achieved with 2 or 3 layers. Along the spectral perspective on GNNs (Bruna et al., 2013; Defferrard et al., 2016; Bronstein et al., 2017; NT & Maehara, 2019), Li et al. (2018) and Wu et al. (2019) explain the failure of deep GCNs by the over-smoothing of the node features. Oono & Suzuki (2020) show an exponential loss of expressive power as the depth in GCNs increases in the sense that the hidden node states tend to converge to Laplacian sub-eigenspaces as the depth increases to infinity. Alon & Yahav (2020) show an over-squashing effect of deep GNNs, in the sense that the width of the hidden states needs to grow exponentially in the depth in order to retain all information about long-range interactions. In comparison, our work focuses on more general GNNs based on neighborhood-aggregation that are not limited in the hidden state widths, and demonstrates the their *advantage* in expressive power compared to GA-MLP models at finite depth, in terms of distinguishing rooted graphs for node-prediction tasks. On the other hand, there exist examples of useful deep GNNs. Chen et al. (2019b) apply 30-layer GNNs for community detection problems, which uses a family of multi-scale operators as well as normalization steps (Ioffe & Szegedy, 2015; Ulyanov et al., 2016). Recently, Li et al. (2019; 2020a) and Chen et al. (2020a) build deeper GCN architectures with the help of various residual connections (He et al., 2016) and normalization steps to achieve impressive results in standard datasets, which further highlights the need to study the role of depth in GNNs. Gong et al. (2020) propose geometrically principled connections, which improve upon vanilla residual connections on graph- and mesh-based tasks.

**Existing GA-MLP-type models** Motivated by better understanding GNNs as well as enhancing computational efficiency, several models of the GA-MLP type have been proposed and they achieve competitive performances on various datasets. Wu et al. (2019) propose the Simple Graph Convolution (SGC), which removes the intermediary weights and nonlinearities in GCNs. Chen et al. (2019a) propose the Graph Feature Network (GFN), which further adds intermediary powers of the normalized adjacency matrix to the operator family and is applied to graph-prediction tasks. NT & Maehara (2019) propose the Graph Filter Neural Networks (gfNN), which enhances the SGC in the final MLP step. Rossi et al. (2020) propose Scalable Inception Graph Neural Networks (SIGNs), which augments the operator family with Personalized-PageRank-based (Klicpera et al., 2018; 2019) and triangle-based (Monti et al., 2018; Chen et al., 2019b) adjacency matrices.

**Expressive Power of GNNs** Xu et al. (2019) and Morris et al. (2019) show that GNNs based on neighborhood-aggregation are no more powerful than the Weisfeiler-Lehman (WL) test for graph isomorphism (Weisfeiler & Leman, 1968), in the sense that these GNNs cannot distinguish between any pair of non-isomorphic graphs that the WL test cannot distinguish. They also propose models that match the expressive power of the WL test. Since then, many attempts have been made to build

GNN models whose expressive power are not limited by WL (Morris et al., 2019; Maron et al., 2019a; Chen et al., 2019c; Morris & Mutzel, 2019; You et al., 2019; Bouritsas et al., 2020; Li et al., 2020b; Flam-Shepherd et al., 2020; Sato et al., 2019; 2020). Other perspectives for understanding the expressive power of GNNs include function approximation (Maron et al., 2019b; Chen et al., 2019c; Keriven & Peyré, 2019), substructure counting (Chen et al., 2020b), Turing universality (Loukas, 2020) and the determination of graph properties (Garg et al., 2020). Sato (2020) provides a survey on these topics. In this paper, besides studying the expressive power of GA-MLPs along the line of graph isomorphism tests, we propose a new perspective of approximating functions on rooted graphs, which is motivated by node-prediction tasks, and show a gap between GA-MLPs and GNNs that grows exponentially in the size of the receptive field in terms of the equivalence classes that they induce on rooted graphs.

## 3 BACKGROUND

### 3.1 NOTATIONS

Let $G = (V, E)$ denote a graph, with $V$ being the vertex set and $E$ being the edge set. Let $n$ denote the number of nodes in $G$, $A \in \mathbb{R}^{n \times n}$ denote the *adjacency matrix*, $D \in \mathbb{R}^{n \times n}$ denote the diagonal *degree matrix* with $D_{ii} = d_i$ being the degree of node $i$. We call $D^{-\frac{1}{2}} A D^{-\frac{1}{2}}$ the *(symmetrically) normalized adjacency matrix*, and $D^{-\alpha} A D^{-\beta}$ a *generalized normalized adjacency matrix* for any $\alpha, \beta \in \mathbb{R}$. Let $X \in \mathbb{R}^{n \times d}$ denote the matrix of node features, where $X_i$ denotes the $d$-dimensional feature that node $i$ possesses. For a node $i \in V$, let $\mathcal{N}(i)$ denote the set of neighbors of $i$. We assume that the edges do not possess features. In a node prediction task, the labels are given by $Y \in \mathbb{R}^n$.

For a positive integer $K$, we let $[K] = \{1, ..., K\}$. We use $\{...\}_m$ to denote a multiset, which allows repeated elements. We say a function $f(K)$ is *doubly-exponential* in $K$ if $\log \log f(K)$ is polynomial in $K$, and *poly-exponential* in $K$ if $\log f(K)$ is polynomial in $K$, as $K$ tends to infinity.

### 3.2 GRAPH NEURAL NETWORKS (GNNs)

Following the notations in Xu et al. (2019), we consider $K$-layer GNNs defined generically as follows. For $k \in [K]$, we compute the hidden node states $H \in \mathbb{R}^{n \times d^{(k)}}$ iteratively as

$$M_i^{(k)} = \text{AGGREGATE}^{(k)}(\{H_j^{(k-1)} : j \in \mathcal{N}(i)\}), \quad H_i^{(k)} = \text{COMBINE}^{(k)}(H_i^{(k-1)}, M_i^{(k)}), \quad (1)$$

where we set $H^{(0)} = X$ to be the node features. If a graph-level output is desired, we finally let

$$Z_G = \text{READOUT}(\{H_i^{(K)} : i \in V\}), \quad (2)$$

Different choices of the trainable COMBINE, AGGREGATE and READOUT functions result in different GNN models, though usually AGGREGATE and READOUT are chosen to be permutation-invariant. As graph-level functions, it is shown in Xu et al. (2019) and Morris et al. (2019) that the maximal expressive power of models of this type coincides with running $K$ iterations of the WL test for graph isomorphism, in the sense that any two non-isomorphic graphs that cannot be distinguished by the latter cannot be distinguished by the $K$-layer GNNs, either. For this reason, we will not distinguish between GNN and WL in discussions on expressive powers.

### 3.3 GRAPH-AUGMENTED MULTI-LAYER PECEPTRONS (GA-MLPs)

GA-MLPs are models that consist of two steps - first augmenting the node features with some operators based on the graph topology, and then applying a node-wise learnable function. Below we focus on using linear graph operators to augment the node features, while an extension of the definition as well as some of the theoretical results in Section 5 to GA-MLPs using general graph operators is given in Appendix A. Let $\Omega = \{\omega_1(A), ..., \omega_K(A)\} \subseteq \mathbb{R}^{n \times n}$ be a set of (usually multi-hop) linear operators that are functions of the adjacency matrix, $A$. Common choices of the operators are powers of the (normalized) adjacency matrix, and several particular choices of $\Omega$ that give rise to existing GA-MLP models are listed in Appendix B. In its general form, a GA-MLP first computes a series of augmented features via

$$\tilde{X}_k = \omega_k(A) \cdot \varphi(X) \in \mathbb{R}^{n \times \tilde{d}}, \quad (3)$$

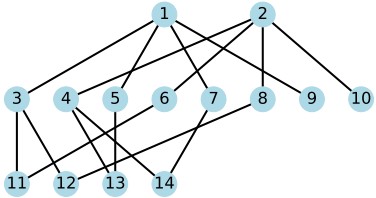 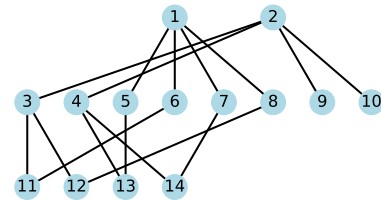

Figure 1: A pair of graphs that can be distinguished by 2 iterations of the WL test but not by GA-MLPs with $\Omega \subseteq \{A^k : k \in \mathbb{N}\}$, as proved in Appendix K.

with $\varphi : \mathbb{R}^d \to \mathbb{R}^{\tilde{d}}$ being a learnable function acting as a feature transformation applied to each node separately. It can be realized by an MLP, e.g. $\varphi(X) = \sigma(XW_1)W_2$, where $\sigma$ is a nonlinear activation function and $W_1, W_2$ are trainable weight matrices of suitable dimensions. Next, the model concatenates $\tilde{X}_1, ..., \tilde{X}_K$ into $\tilde{X} = [\tilde{X}_1, ..., \tilde{X}_K] \in \mathbb{R}^{n \times (K\tilde{d})}$, and computes

$$Z = \rho(\tilde{X}) \in \mathbb{R}^{n \times d'} , \tag{4}$$

where $\rho : \mathbb{R}^{K\tilde{d}} \to \mathbb{R}^{d'}$ is also a learnable node-wise function, again usually realized by an MLP. If a graph-level output is desired, we can also add a READOUT function as in (2).

A simplified version of the model sets $\varphi$ to be the identity function, in which case (3) and (4) can be written together as

$$Z = \rho([\omega_1(A) \cdot X, ..., \omega_K(A) \cdot X]) \tag{5}$$

Such a simplification improves computational efficiency since the matrix products $\omega_k(A) \cdot X$ can be pre-computed before training (Rossi et al., 2020). Since we are mostly interested in an upper bounds on the expressive power of GA-MLPs, we will work with the more general update rule (3) in this paper, but the lower-bound result in Proposition 2 remains valid even when we restrict to the subset of models where $\varphi$ is taken to be the identity function.

## 4 EXPRESSIVE POWER AS GRAPH ISOMORPHISM TESTS

We first study the expressive power of GA-MLPs via their ability to distinguish non-isomorphic graphs. It is not hard to see that when $\Omega = \{I, \tilde{A}, ..., \tilde{A}^K\}$, where $\tilde{A} = D^{-\alpha}AD^{-\beta}$ for any $\alpha, \beta \in \mathbb{R}$ generalizes the normalized adjacency matrix, this is upper-bounded by the power of $K + 1$ iterations of WL. We next ask whether it can fall strictly below. Indeed, for two common choices of $\Omega$, we can find concrete examples: 1) If $\Omega$ consists of integer powers of any normalized adjacency matrix of the form $D^{-\alpha}AD^{-(1-\alpha)}$ for some $\alpha \in [0, 1]$, then it is apparent that the GA-MLP cannot distinguish any pair of *regular graphs* with the same size but different node degrees; 2) If $\Omega$ consists of integer powers of the adjacency matrix, $A$, then the model cannot distinguish between the pair of graphs shown in Figure 1, which can be distinguished by 2 iterations of the WL test. The proof of the latter result is given in Appendix K. Together, we summarize the results as:

**Proposition 1.** *If $\Omega \subseteq \{\tilde{A}^k : k \in \mathbb{N}\}$, with either $\tilde{A} = A$ or $\tilde{A} = D^{-\alpha}AD^{-(1-\alpha)}$ for some $\alpha \in [0, 1]$, there exists a pair of graphs which can be distinguished by GNNs but not this GA-MLP.*

Nonetheless, if we focus on not particular counterexamples but rather the average performance in distinguishing random graphs, it is not hard for GA-MLPs to reach the same level as WL, which is known to distinguish almost all pairs of random graphs under a uniform distribution (Babai et al., 1980). Specifically, building on the results in Babai et al. (1980), we prove in Appendix F that:

**Proposition 2.** *For all $n \in \mathbb{N}_+$, $\exists \alpha_n > 0$ such that any GA-MLP that has $\{D, AD^{-\alpha_n}\} \subseteq \Omega$ can distinguish almost all pairs of non-isomorphic graphs of at most $n$ nodes, in the sense that the fraction of graphs on which such a GA-MLP fails to test isomorphism is $1 - o(1)$ as $n \to \infty$.*

The hypothesis that distinguishing non-isomorphic graphs is not difficult on average for either GNNs or GA-MLPs is further supported by the numerical results provided in Appendix C, in which we count the number of equivalence classes that either of them induce on graphs that occur in real-world datasets. This further raises the question of whether graph isomorphism tests along suffice as a

criterion for comparing the expressive power of models on graphs, which leads us to the explorations in the next section.

Lastly, we remark that with suitable choices of operators in $\Omega$, it is possible for GA-MLPs to go beyond the power of WL. For example, if $\Omega$ contains the *power graph adjacency matrix* introduced in Chen et al. (2019b), $\min(A^2, 1)$, then the GA-MLP can distinguish between a hexagon and a pair of triangles, which WL cannot distinguish.

## 5 EXPRESSIVE POWER AS FUNCTIONS ON ROOTED GRAPHS

To study the expressive power beyond graph isomorphism tests, we consider the setting of node-wise prediction tasks, for which the final readout step (2) is dropped in both GNNs and GA-MLPs. Whether the learning setup is transductive or inductive, we can consider the models as functions on *rooted graphs*, or *egonets* (Preciado & Jadbabaie, 2010), which are graphs with one node designated as the root $\{i_1, ..., i_n\}$ is a set of nodes in the graphs $\{G_1, ..., G_n\}$ (not necessarily distinct) and with node-level labels $\{Y_{i_1}, ..., Y_{i_n}\}$ known during training, respectively, then the goal is to fit a function to the input-output pairs $(G_n^{[i_n]}, Y_{i_n})$, where we use $G^{[i]}$ to denote the rooted graph with $G$ being the graph and the node $i$ in $G$ being the root. Thus, we can evaluate the expressive power of GNNs and GA-MLPs by their ability to approximate functions on the space of rooted graphs, which we call $\mathcal{E}$.

To do so, we introduce a notion of induced equivalence relations on $\mathcal{E}$, analogous to the equivalence relations on $\mathcal{G}$ introduced in Appendix C. Given a family of functions $\mathcal{F}$ on $\mathcal{E}$, we can define an equivalence relation $\simeq_{\mathcal{E};\mathcal{F}}$ among all rooted graphs such that $\forall G^{[i]}, G'^{[i']} \in \mathcal{E}$, $G^{[i]} \simeq_{\mathcal{E};\mathcal{F}} G'^{[i']}$ if and only if $\forall f \in \mathcal{F}$, $f(G^{[i]}) = f(G'^{[i']})$. By examining the number and sizes of the induced equivalence classes of rooted graphs, we can evaluate the relative expressive power of different families of functions on $\mathcal{E}$ in a quantitative way.

In the rest of this section, we assume that the node features belong to a finite alphabet $\mathcal{X} \subseteq \mathbb{N}$ and all nodes have degree at most $m \in \mathbb{N}_+$. Firstly, GNNs are known to distinguish neighborhoods up to the *rooted aggregation tree*, which can be obtained by unrolling the neighborhood aggregation steps in the GNNs as well as the WL test (Xu et al., 2019; Morris et al., 2019; Garg et al., 2020). The *depth-K rooted aggregation tree* of a rooted graph $G^{[i]}$ is a depth-$K$ rooted tree with a (possibly many-to-one) mapping from every node in the tree to some node in $G^{[i]}$, where (i) the root of the tree is mapped to node $i$, and (ii) the children of every node $j$ in the tree are mapped to the neighbors of the node in $G^{[i]}$ to which $j$ is mapped. An illustration of rooted graphs and rooted aggregation trees is given in Figure 4. Hence, each equivalence class in $\mathcal{E}$ induced by the family of all depth-$K$ GNNs consists of all rooted graphs that share the same rooted aggregation tree of depth-$K$. Thus, to estimate the number of equivalence classes on $\mathcal{E}$ induced by GNNs, we need to estimate the number of possible rooted aggregation trees, which is given by Lemma 3 in Appendix G. Thus, we derive the following lower bound on the number of equivalence classes in $\mathcal{E}$ that depth-$K$ GNNs induce:

**Proposition 3.** *Assume that $|\mathcal{X}| \geq 2$ and $m \geq 3$. The total number of equivalence classes of rooted graphs induced by GNNs of depth $K$ grows at least doubly-exponentially in $K$.*

In comparison, we next demonstrate that the equivalence classes induced by GA-MLPs are more coarsened. To see this, let's first consider the example where we take $\Omega = \{I, \tilde{A}, \tilde{A}^2, ..., \tilde{A}^K\}$, in which $\tilde{A} = D^{-\alpha} A D^{-\beta}$ with any $\alpha, \beta \in \mathbb{R}$ is a generalization of the normalized adjacency matrix. From formula (3), by expanding the matrix product, we have

$$(\tilde{A}^k \varphi(X))_i = \sum_{(i_1,...,i_k) \in \mathcal{W}_k(G^{[i]})} d_i^{-\alpha} d_{i_1}^{-(\alpha+\beta)} ... d_{i_{k-1}}^{-(\alpha+\beta)} d_{i_k}^{-\beta} \varphi(X_{i_k}) , \tag{6}$$

where we define $\mathcal{W}_k(G^{[i]}) = \{(i_1, ..., i_k) \subseteq V : A_{i,i_1}, A_{i_1,i_2}, ..., A_{i_{k-1},i_k} > 0\}$ to be set of all *walks* of length $k$ in the rooted graph $G^{[i]}$ starting from node $i$ (an illustration is given in Figure 2). Thus, the $k$th augmented feature of node $i$, $(\tilde{A}^k \varphi(X))_i$, is completely determined by the number of each "type" of walks in $G^{[i]}$ of length $k$, where the type of a walk, $(i_1, ..., i_k)$, is determined jointly by the degree multiset, $\{d_{i_1}, ..., d_{i_{k-1}}\}$ as well as the degree and the node feature of the end node, $d_{i_k}$ and $X_{i_k}$. Hence, to prove an upper bound on the total number of equivalence classes on $\mathcal{E}$ induced by such a GA-MLP, it is sufficient to upper-bound the total number of possibilities of

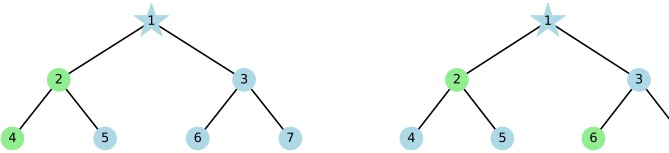

Figure 2: A pair of rooted graphs, $G^{[1]}$ (left) and $G'^{[1]}$ (right), in which blue nodes have node feature 0 and green nodes have node feature 1. They belong to the same equivalence class induced by any GA-MLP with operators that only depend on the graph structure, but different equivalence classes induced by GNNs. In particular, $G^{[1]}$ and $G'^{[1]} \in \mathcal{T}_{2,2,(1,1,3)}$ (defined in Appendix I), and $|\mathcal{W}_2(G^{[1]};(1,1))| = 1$ whereas $|\mathcal{W}_2(G'^{[1]};(1,1))| = 0$.

assigning the counts of all types of walks in a rooted graph. This allows us to derive the following result, which we prove in Appendix H.

**Proposition 4.** *Fix* $\Omega = \{I, \tilde{A}, \tilde{A}^2, ..., \tilde{A}^K\}$, *where* $\tilde{A} = D^{-\alpha}AD^{-\beta}$ *for some* $\alpha, \beta \in \mathbb{R}$. *Then the total number of equivalence classes in* $\mathcal{E}$ *induced by such GA-MLPs is poly-exponential in* $K$.

Compared with Proposition 3, this shows that the number of equivalence classes on $\mathcal{E}$ induced by such GA-MLPs is exponentially smaller than that by GNNs. In addition, as the other side of the same coin, these results also indicate the complexity of these hypothesis classes. Building on the results in Chen et al. (2019c; 2020b) on the equivalence between distinguishing non-isomorphic graphs and approximating arbitrary permutation-invariant functions on graphs by neural networks, and by the definition of *VC dimension* (Vapnik & Chervonenkis, 1971; Mohri et al., 2018), we conclude that

**Corollary 1.** *The VC dimension of all GNNs of* $K$ *layers as functions on rooted graphs grows at least doubly-exponentially in* $K$; *Fixing* $\alpha, \beta \in \mathbb{R}$, *the VC dimension of all GA-MLPs with* $\Omega = \{I, \tilde{A}, \tilde{A}^2, ..., \tilde{A}^K\}$ *as functions on rooted graphs is at most poly-exponential in* $K$.

Meanwhile, for more general operators, we can show that the equivalence classes induced by GA-MLPs are *coarser* than those induced by GNNs at least under some measurements. For instance, the pair of rooted graphs in Figure 2 belong to the same equivalence class induced by any GA-MLP (as we prove in Appendix I) but different equivalence classes induced by GNNs. Rigorously, we characterize such a gap in expressive power by finding certain equivalence classes in $\mathcal{E}$ induced by GA-MLPs that intersect with many equivalence classes induced by GNNs. In particular, we have the following general result, which we prove in Appendix I:

**Proposition 5.** *If* $\Omega$ *is any family of equivariant linear operators on the graph that only depend on the graph topology of at most* $K$ *hops, then there exist exponentially-in-*$K$ *many equivalence classes in* $\mathcal{E}$ *induced by the GA-MLPs with* $\Omega$, *each of which intersects with doubly-exponentially-in-*$K$ *many equivalence classes in* $\mathcal{E}$ *induced by depth-*$K$ *GNNs, assuming that* $|\mathcal{X}| \geq 2$ *and* $m \geq 3$. *Conversely, in constrast, if* $\Omega = \{I, \tilde{A}, \tilde{A}^2, ..., \tilde{A}^K\}$, *in which* $\tilde{A} = D^{-\alpha}AD^{-\beta}$ *with any* $\alpha, \beta \in \mathbb{R}$, *then each equivalence class in* $\mathcal{E}$ *induced by depth-*$(K+1)$ *GNNs is contained in one equivalence class induced by the GA-MLPs with* $\Omega$.

In essence, this result establishes that GA-MLP circuits can express fewer (exponentially fewer) functions than GNNs with equivalent receptive field. Taking a step further, we can find explicit functions on rooted graphs that can be approximated by GNNs but not GA-MLPs. In the framework that we have developed so far, this occurs when the image of each equivalence class in $\mathcal{E}$ induced by GNNs under this function contains a single value, whereas the image of some equivalence class in $\mathcal{E}$ induced by GA-MLPs contains multiple values. Inspired by the proofs of the results above, a natural candidate is the family of functions that count the number of walks of a particular type in the rooted graph. We can establish the following result, which we prove in Appendix J:

**Proposition 6.** *For any sequence of node features* $\{x_k\}_{k \in \mathbb{N}_+} \subseteq \mathcal{X}$, *consider the sequence of functions* $f_k(G^{[i]}) := |\mathcal{W}_k(G^{[i]};(x_1,...,x_k))|$ *on* $\mathcal{E}$. *For all* $k \in \mathbb{N}_+$, *the image under* $f_k$ *of every equivalence class in* $\mathcal{E}$ *induced by depth-*$k$ *GNNs contains a single value, while for any GA-MLP using equivariant linear operators that only depend on the graph topology, there exist exponentially-in-*$k$ *many equivalence classes in* $\mathcal{E}$ *induced by this GA-MLP whose image under* $f_k$ *contains exponentially-in-*$k$ *many values.*

In other words, there exist graph instances where the attributed-walk-counting-function $f_k$ takes different values, yet no GA-MLP model can predict them apart – and there are exponentially many

| # Nodes | Cora | | Citeseer | | Pubmed | |
|---|---|---|---|---|---|---|
| | 2708 | | 3327 | | 19717 | |
| $K$ | GNN | GA-MLP | GNN | GA-MLP | GNN | GA-MLP |
| 1 | 37 | 37 | 31 | 31 | 82 | 82 |
| 2 | 1589 | 756 | 984 | 506 | 8059 | 3762 |
| 3 | 2301 | 2158 | 1855 | 1550 | 12814 | 12014 |
| 4 | 2363 | 2359 | 2074 | 2019 | 12990 | 12979 |
| 5 | 2365 | 2365 | 2122 | 2115 | 12998 | 12998 |

Table 1: The number of equivalence classes of rooted graphs induced by GNN and GA-MLP on node classification datasets with node features removed.

| Model | Cora | | RRG | |
|---|---|---|---|---|
| | Train | Test | Train | Test |
| GIN | 3.98E-6 | 9.72E-7 | 3.39E-5 | 2.61E-4 |
| GA-MLP-$A$ | 1.23E-1 | 1.56E-1 | 1.75E-2 | 2.13E-2 |
| GA-MLP-$A$+ | 1.87E-2 | 6.44E-2 | 1.69E-2 | 2.13E-2 |
| GA-MLP-$\tilde{A}_{(1)}$ | 4.22E-1 | 5.79E-1 | 1.02E-1 | 1.58E-1 |
| GA-MLP-$\tilde{A}_{(1)}$+ | 4.00E-1 | 5.79E-1 | 1.12E-1 | 1.52E-1 |

Table 2: MSE loss divided by label variance for counting attributed walks on the Cora graph and RRG. The models denoted as "+" contain twice as many powers of the operator.

of these instances as the number of hops increases. This suggests the possibility of lower-bounding the average approximation error for certain functions by GA-MLPs under various random graph families, which we leave for future work.

## 6 EXPERIMENTS

The baseline GA-MLP models we consider has operator family $\Omega = \{I, A, ..., A^K\}$ for a certain $K$, and we call it GA-MLP-$A$. In Section 6.2 and 6.3, we also consider GA-MLPs with $\Omega = \{I, \tilde{A}_{(1)}, ..., \tilde{A}_{(1)}^K\}$ ($\tilde{A}_{(\epsilon)}$ is defined in Appendix B), denoted as GA-MLP-$\tilde{A}_{(1)}$. For the experiments in Section 6.3, due to the large $K$ as well as the analogy with spectral methods (Chen et al., 2019b), we use instance normalization (Ulyanov et al., 2016). Further details are described in Appendix L.

### 6.1 NUMBER OF EQUIVALENCE CLASSES OF ROOTED GRAPHS

Motivated by Propositions 3 and 4, we numerically count the number of equivalence classes induced by GNNs and GA-MLPs among the rooted graphs found in actual graphs with node features removed. For depth-$K$ GNNs, we implement a WL-like process with hash functions to map the depth-$K$ egonet associated with each node to a string before comparing across nodes. For GA-MLP-$A$, we compare the augmented features of each egonet computed via (3). From the results in Table 1, we see that indeed, the number of equivalence classes induced by GA-MLP-$A$ is smaller than that by GNNs, with the highest relative difference occurring at $K = 2$. The contrast is much more visible than their difference in the number of *graph* equivalence classes given in Appendix C.

### 6.2 COUNTING ATTRIBUTED WALKS

Motivated by Proposition 6, we test the ability of GNNs and GA-MLPs to count the number of walks of a particular type in synthetic data. We take graphs from the Cora dataset (with node features removed) as well as generate a random regular graph (RRG) with 1000 nodes and the node degree being 6. We assign node feature *blue* to all nodes with even index and node feature *red* to all nodes with odd index, due to which the node feature is given by 2-dimensional one-hot encoding. On the Cora graph, a node $i$'s label is given by the number of walks of the type *blue→blue→blue* starting from $i$. On the RRG, the label is given by the number of walks of the type *blue→blue→blue→blue* starting from $i$. The number of nodes for training and testing is split as $1000/1708$ for the Cora graph and $300/700$ for the random regular graph. We test four GA-MLP models, two with as many powers of the operator as the walk length and the other two with twice as many operators, and compare their performances against that of the Graph Isomorphism Network (GIN), a GNN model that achieves the expressive power of the WL test (Xu et al., 2019). From Table 2, we see that GIN significantly outperforms GA-MLPs in both training and testing on both graphs, consistent with the theoretical result in Proposition 6 that GNNs can count attributed walks while GA-MLPs cannot. Thus, this points out an intuitive task that lies in the gap of expressive power between GNNs and GA-MLPs.

### 6.3 COMMUNITY DETECTION ON STOCHASTIC BLOCK MODELS (SBM)

We use the task of community detection to illustrate another limitation of GA-MLP models: a lack of flexibility to *learn* the family of operators. SBM is a random graph model in which nodes are partitioned into underlying communities and each edge is drawn independently with a probability

that only depends on whether the pair of nodes belong to the same community or not. The task of community detection is then to recover the community assignments from the connectivity pattern. We focus on binary (that is, having two underlying communities) SBM in the sparse regime, where it is known that the hardness of detecting communities is characterized by a signal-to-noise ratio (SNR) that is a function of the in-group and out-group connectivity (Abbe, 2017). We select 5 pairs of in-group and out-group connectivity, resulting in 5 different hardness levels of the task.

Among all different approaches to community detection, spectral methods are particularly worth mentioning here, which usually aim at finding a certain eigenvector of a certain operator that is correlated with the community assignment, such as the second largest eigenvector of the adjacency matrix or the second smallest eigenvector of the Laplacian matrix or the Bethe Hessian matrix (Krzakala et al., 2013). In particular, the Bethe Hessian matrix is known to be asymptotically optimal in the hard regime, provided that a data-dependent parameter is known. Note that spectral methods bear close resemblance to GA-MLPs and GNNs. In particular, Chen et al. (2019b) propose a spectral GNN (sGNN) model for community detection that can be viewed as a learnable generalization of power iterations on a collection of operators. Further details on Bethe Hessian and sGNN are provided in Appendix L.

We first compare two variants of GA-MLP models: GA-MLP-$A$ with $K = 120$, and GA-MLP-$H$ with $\Omega$ generated from the Bethe Hessian matrix with the oracle data-dependent parameter also up to $K = 120$. From Figure 3, we see that the latter consistently outperforms the former, indicating the importance of the choice of the operators for GA-MLPs. As reported in Appendix L, replacing $A$ by $\tilde{A}_{(1)}$ yields no improvement in performance. Meanwhile, we also test a variant of sGNN that is only based on powers of the $A$ and has the same receptive field as GA-MLP-$A$ (further details given in Appendix L). We see that its performance is comparable to that of GA-MLP-$H$. Thus, this

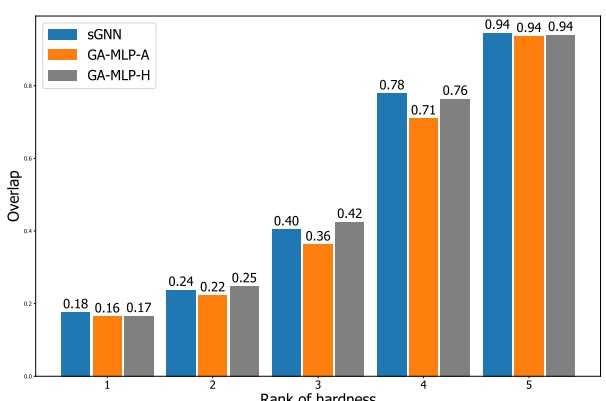

Figure 3: Community detection on binary SBM with 5 choices of in- and out-group connectivities, each yielding to a different SNR. Higher overlap means better performance.

demonstrates a scenario in which GA-MLP with common choices of $\Omega$ do not work well, but there exists some choice of $\Omega$ that is a priori unknown, with which GA-MLP can achieve good performance. In contrast, a GNN model does not need to rely on the knowledge of such an oracle set of operators, demonstrating its superior capability of learning.

## 7 CONCLUSIONS

We have studied the separation in terms of representation power between GNNs and a popular alternative that we coined GA-MLPs. This latter family is appealing due to its computational scalability and its conceptual simplicity, whereby the role of topology is reduced to creating 'augmented' node features then fed into a generic MLP. Our results show that while GA-MLPs can distinguish almost all non-isomorphic graphs, in terms of approximating node-level functions, there exists a gap growing exponentially-in-depth between GA-MLPs and GNNs in terms of the number of equivalence classes of nodes (or rooted graphs) they induce. Furthermore, we find a concrete class of functions that lie in this gap given by the counting of attributed walks. Moreover, through community detection, we demonstrate the lack of GA-MLP's ability to go beyond the fixed family of operators as compared to GNNs. In other words, GNNs possess an inherent ability to discover topological features through learnt diffusion operators, while GA-MLPs are limited to a fixed family of diffusions.

While we do not attempt to provide a decisive answer of whether GNNs or GA-MLPs should be preferred in practice, our theoretical framework and concrete examples help to understand their differences in expressive power and indicate the types of tasks in which a gap is more likely to

be seen – those exploiting stronger structures among nodes like counting attributed walks, or those involving the learning of graph operators. That said, our results are purely on the representation side, and disregard optimization considerations; integrating the possible optimization counterparts is an important direction of future improvement. Finally, another open question is to better understand the links between GA-MLPs and spectral methods, and how this can help learning diffusion operators.

## ACKNOWLEDGEMENTS

We are grateful to Jiaxuan You for initiating the discussion on GA-MLP-type models, as well as Mufei Li, Minjie Wang, Xiang Song, Lingfan Yu, Michael M. Bronstein and Shunwang Gong for helpful conversations. This work is partially supported by the Alfred P. Sloan Foundation, NSF RI-1816753, NSF CAREER CIF 1845360, NSF CHS-1901091, Capital One, Samsung Electronics, and the Institute for Advanced Study.

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

# A  GA-MLP WITH GENERAL EQUIVARIANT GRAPH OPERATORS FOR NODE FEATURE AUGMENTATION

For a graph $G = (V, E)$ with $n$ nodes, assume without loss of generality that $V = [n]$. Let $\mathbb{S}_n$ denote the set of permutations of $n$, and $\forall \pi \in \mathbb{S}_n$, it maps a node $i \in [n]$ to $\pi(i) \in [n]$. For $\pi \in \mathbb{S}_n$ and a matrix $M \in \mathbb{R}^{n \times n}$, we use $\pi \star M \in \mathbb{R}^{n \times n}$ to denote the $\pi$-permuted version of $M$, that is, $(\pi \star M)_{i,j} = M_{\pi(i),\pi(j)}$. For $\pi \in \mathbb{S}_n$ and a matrix $Z \in \mathbb{R}^{n \times d}$, we use $\pi \star Z \in \mathbb{R}^{n \times d}$ to denote the $\pi$-permuted version of $M$, that is, $(\pi \star Z)_{i,p} = Z_{\pi(i),p}$.

Below, we define a more general form of GA-MLP models that extend the use of equivariant linear operators for node feature propagation to that of general equivariant graph operators. We first define a map $\omega : \mathbb{R}^{n \times n} \times \mathbb{R}^{n \times d} \to \mathbb{R}^{n \times d'}$, whose first input argument is always the adjacency matrix of a graph, $A$, and second input argument is a node feature matrix. We say the map satisfies *equivariance* to node permutations if $\forall \pi \in \mathbb{S}_n$, $\forall Z \in \mathbb{R}^{n \times d}$, there is $\omega(\pi \star A, \pi \star Z) = \pi \star \omega(A, Z)$. With a slight abuse of notations, we also use $\omega[A](Z)$ to denote $\omega(A, Z)$, thereby considering $\omega[A] : \mathbb{R}^{n \times d} \to \mathbb{R}^{n \times d'}$ as an operator on node features. If $\omega$ satisfies equivariance to node permutations as defined above, we then call $\omega[A]$ an equivariant graph operator. We can then define a general (nonlinear) GA-MLP model as

$$\tilde{X} = \omega[A](X)$$
$$Z = \rho(\tilde{X}) \tag{7}$$

where $\omega$ is an equivariant graph operator, and $\rho$ is a node-wise function.

It is easy to see that

**Proposition 7.** *If $\omega[A](X) = m(A) \cdot X$, where $m(\cdot) = \mathbb{R}^{n \times n} \to \mathbb{R}^{n \times n}$ is an entry-wise function or matrix product or compositions thereof, then $\omega[A]$ is an equivariant graph operator.*

## A.1  EXTENDING THE PROOF OF PROPOSITION 5 AND 6 TO GENERAL GA-MLPS

An extension of the first half of Proposition 5 is

**Proposition 8.** *If $\omega[A]$ is an equivariant graph operator, then there exist exponentially-in-$K$ many equivalence classes in $\mathcal{E}$ induced by the general GA-MLPs with $\omega[A]$, each of which intersects with doubly-exponentially-in-$K$ many equivalence classes in $\mathcal{E}$ induced by depth-$K$ GNNs, assuming that $|\mathcal{X}| \geq 2$ and $m \geq 3$.*

*Proof*: Similar to the proof of Proposition 5 given in Appendix I, we consider the set of full $m$-ary rooted trees of depth $K$, $\mathcal{T}_{m,K,\mathcal{X}}$, that is all rooted trees of depth $K$ in which the nodes have features belonging to the discrete set $\mathcal{X} \subseteq \mathbb{N}$ and all non-leaf nodes have $m$ children. $\mathcal{T}_{m,K,\mathcal{X}}$ is a subset of $\mathcal{E}$, the space of all rooted graphs. Suppose $f$ is a function represented by a general GA-MLP defined in (7) with an equivariant graph operator $\omega[A]$. Let $V_k$ denote the set of nodes at depth $k$ of $T$. Notice the following symmetry among nodes in each $V_k$: if $\pi$ is the permutation of a pair of nodes in some $V_k$ for $1 \leq k \leq K$, then $\pi \star A = A$. By the equivariance property of $\omega$, this implies that

$$\omega[A](\pi \star Z) = \omega[\pi \star A](\pi \star Z)$$
$$= \pi \star \omega[A](Z) \tag{8}$$

Let $X$ denote the node feature matrix associated with $T$, and $\pi \star T$ denote the rooted tree in $\mathcal{T}_{m,K,\mathcal{X}}$ with the same topology (i.e., also a full $m$-ary rooted tree) but node feature matrix $\pi \star X$. Then, since the root node is not permuted under $\pi$, we know that

$$f(T) = \rho\left(\omega[A](X)_{1,:}\right)$$
$$= \rho\left((\pi \star \omega[A](X))_{1,:}\right)$$
$$= \rho\left(\omega[A](\pi \star X)_{1,:}\right) \tag{9}$$
$$= f(\pi \star T)$$

This implies that for two trees $T$ and $T' \in \mathcal{T}_{m,K,\mathcal{X}}$, if $\forall 0 \leq k \leq K, \forall x \in \mathcal{X}$, they satisfy $|\overline{\mathcal{W}}_k(T; x)| = |\overline{\mathcal{W}}_k(T'; x)|$, then $f(T) = f(T')$ for all such $f$'s, and hence $T$ and $T'$ belong to

the same equivalence class in $\mathcal{E}$ induced by GA-MLPs. Therefore, by the rest of the argument given in Proposition 5, Proposition 8 can be proven analogously for GA-MLPs with general equivariant graph operators. □

Similarly, Proposition 6 can also be extended to

**Proposition 9.** *For any sequence of node features $\{x_k\}_{k \in \mathbb{N}_+} \subseteq \mathcal{X}$, consider the sequence of functions $f_k(G^{[i]}) := |\mathcal{W}_k(G^{[i]}; (x_1, ..., x_k))|$ on $\mathcal{E}$. For all $k \in \mathbb{N}_+$, the image under $f_k$ of every equivalence class in $\mathcal{E}$ induced by depth-$k$ GNNs contains a single value, while for any GA-MLP using equivariant graph operators, there exist exponentially-in-$k$ many equivalence classes in $\mathcal{E}$ induced by this GA-MLP whose image under $f_k$ contains exponentially-in-$k$ many values.*

The proof replies on the same extension as described above in the proof of Proposition 8.

# B  EXAMPLES OF EXISTING GA-MLP MODELS

For $\epsilon \in \mathbb{R}$, let $\bar{A}_{(\epsilon)} = A + \epsilon I$, $\bar{D}_{(\epsilon)}$ be the diagonal matrix with $\bar{D}_{(\epsilon),ii} = \sum_j A_{ij} + \epsilon$, and $\tilde{A}_{(\epsilon)} = \bar{D}_{(\epsilon)}^{-1/2} \bar{A}_{(\epsilon)} \bar{D}_{(\epsilon)}^{-1/2}$.

- Simple Graph Convolution (Wu et al., 2019):
  $\Omega(A) = \{(\tilde{A}_{(1)})^K\}$ for some $K > 0$. In addition, $\varphi$ is the identity function and $\rho(H) = softmax(HW)$ for some trainable weight matrix $W$.
- Graph Feature Network (Chen et al., 2019a):
  $\Omega(A) = \{I, D, \tilde{A}_{(\epsilon)}, ..., (\tilde{A}_{(\epsilon)})^K\}$ for some $K > 0$ and $\epsilon > 0$. In addition, $\varphi$ is the identity function and $\rho$ is an MLP.
- Scalable Inception Graph Networks (Rossi et al., 2020):
  $\Omega(A) = \{I\} \cup \Omega_1(A) \cup \Omega_2(A) \cup \Omega_3(A)$, where $\Omega_1(A)$ is a family of simple / normalized adjacency matrices, $\Omega_2(A)$ is a family of Personalized-PageRank-based adjacency matrices, and $\Omega_3(A)$ is a family of triangle-based adjacency matrices. In addition, writing $\tilde{X} = [\tilde{X}_1, ..., \tilde{X}_K]$, there is $Z = \rho(\tilde{X}) = \sigma_1(\sigma_2([\tilde{X}_1 W_1, ..., \tilde{X}_K W_K]) W_{\text{out}})$, with $\sigma_1$ and $\sigma_2$ being nonlinear activation functions and $W_1, ..., W_K$ and $W_{\text{out}}$ being trainable weight matrices of suitable dimensions.

# C  EQUIVALENCE CLASSES INDUCED BY GNNS AND GA-MLPS AMONG REAL GRAPHS

| | IMDBBINARY | | IMDBMULTI | | REDDITBINARY | | REDDITMULTI5K | | COLLAB | |
|---|---|---|---|---|---|---|---|---|---|---|
| # Graphs | 1000 | | 1500 | | 2000 | | 5000 | | 5000 | |
| $K$ | GNN | GA-MLP | GNN | GA-MLP | GNN | GA-MLP | GNN | GA-MLP | GNN | GA-MLP |
| 1 | 51 | 51 | 49 | 49 | 781 | 781 | 1365 | 1365 | 294 | 294 |
| 2 | 537 | 537 | 387 | 387 | 1998 | 1998 | 4999 | 4999 | 4080 | 4080 |
| 3 | 537 | 537 | 387 | 387 | 1998 | 1998 | 4999 | 4999 | 4080 | 4080 |
| *ground truth* | 537 | | 387 | | 1998 | | 4999 | | 4080 | |

Table 3: The number of equivalence classes of graphs induced by GNN and GA-MLP on real datasets with node features removed. The last row gives the ground-truth number of isomorphism classes of graphs computed from the implementation of Ivanov et al. (2019).

Given a space of graphs, $\mathcal{G}$, and a family $\mathcal{F}$ of functions mapping $\mathcal{G}$ to $\mathbb{R}$, $\mathcal{F}$ induces an equivalence relation that we denote by $\simeq_{\mathcal{G};\mathcal{F}}$ among graphs in $\mathcal{G}$ such that for $G_1, G_2 \in \mathcal{G}$, $G_1 \simeq_{\mathcal{G};\mathcal{F}} G_2$ if and only if $\forall f \in \mathcal{F}$, $f(G_1) = f(G_2)$. For example, if $\mathcal{F}$ is powerful enough to distinguish all pairs of non-isomorphic graphs, then each equivalence class under $\simeq_{\mathcal{G},\mathcal{F}}$ contains exactly one graph. Thus, by examining the number or sizes of the equivalence classes induced by different families of functions on $\mathcal{G}$, we can evaluate their relative expressive power in a quantitative way.

Hence, we supplement the theoretical result of Proposition 2 with the following numerical results on five real-world datasets for graph-predictions. For graphs in each of the two real datasets, we

remove their node features and count the total number of equivalence classes among them induced by depth-$K$ GNNs (equivalent to $K$-iterations of the WL test, as discussed in Section 3.2) as well as GA-MLPs with $\Omega = \{I, A, ..., A^K\}$ for different $K$'s. We see from the results in Table 3 that as soon as $K \geq 2$, the number of equivalence classes induced by GNNs and the GA-MLPs are both close to the total number of graphs up to isomorphism, implying that they are indeed both able to distinguish almost all pairs of non-isomorphic graphs among the ones occurring in these datasets.

## D  ADDITIONAL NOTATIONS

For any $k \in \mathbb{N}_+$ and any rooted graph $G^{[i]} = (V, E, i) \in \mathcal{E}$, define

$$\mathcal{W}_k(G^{[i]}) = \{(i_1, ..., i_k) \subseteq V : A_{i,i_1}, A_{i_1,i_2}, ..., A_{i_{k-1},i_k} > 0\} \tag{10}$$

$$\overline{\mathcal{W}}_k(G^{[i]}) = \{(i_1, ..., i_k) \in \mathcal{W}_k(G^{[i]}) : i \neq i_2, i_1 \neq i_3, ..., i_{k-3} \neq i_{k-1}, i_{k-2} \neq i_k\} \tag{11}$$

as the sets of *walks* and *non-backtracking walks* of length $k$ in $G^{[i]}$ starting from the root node, respectively. Note that when $G^{[i]}$ is a rooted tree, a non-backtracking walk of length $k$ is a path from the root node to a node at depth $k$. In addition, for $0 \leq d_1, ..., d_k \leq m$ and $x_1, ..., x_k \in \mathcal{X}$, define the following subsets of $\mathcal{W}_k(G^{[i]})$:

$$\mathcal{W}_k\left(G^{[i]}; (d_1, ..., d_k), x_k\right) = \{(i_1, ..., i_k) \in \mathcal{W}_k(G^{[i]}) : \{d_{i_1}, ..., d_{i_k}\}_m = \{d_1, ..., d_k\}_m, X_{i_k} = x_k\} \tag{12}$$

$$\mathcal{W}_k\left(G^{[i]}; (x_1, ..., x_k)\right) = \{(i_1, ..., i_k) \in \mathcal{W}_k(G^{[i]}) : (X_{i_1}, ..., X_{i_k}) = (x_1, ..., x_k)\} \tag{13}$$

$$\mathcal{W}_k(G^{[i]}; x_k) = \{(i_1, ..., i_k) \in \mathcal{W}_k(G^{[i]}) : X_{i_k} = x_k\} \tag{14}$$

We also define $\overline{\mathcal{W}}_k\left(G^{[i]}; (d_1, ..., d_k), x_k\right)$, $\overline{\mathcal{W}}_k\left(G^{[i]}; (x_1, ..., x_k)\right)$ and $\overline{\mathcal{W}}_k(G^{[i]}; x_k)$ similarly.

## E  ILLUSTRATION OF ROOTED GRAPHS AND ROOTED AGGREGATION TREES

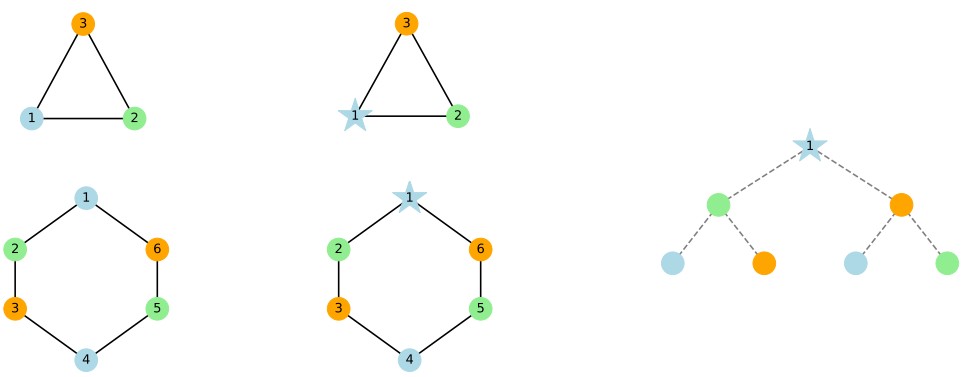

Figure 4: An illustration of rooted graphs and rooted aggregation trees. *Left*: a pair of graphs, $G$ and $G'$. *Center*: the rooted graphs of 1 in $G$ and $G'$, $G^{[1]}$ and $G'^{[1]}$. *Right*: the rooted aggregation tree that both $G^{[1]}$ and $G'^{[1]}$ correspond to.

## F  PROOF OF PROPOSITION 2

With node features being identical in the random graphs, we take $X \in \mathbb{R}^{n \times 1}$ to be the all-1 vector. Thus,

$$(DX)_i = d_i, \tag{15}$$

and

$$(AD^{-\alpha}X)_i = \sum_{j \in \mathcal{N}(i)} d_j^{-\alpha} \,. \tag{16}$$

Since (4) and (2) together can approximate arbitrary permutation-invariant functions on multisets (Zaheer et al., 2017), if two graphs $G = (V, E)$ and $G' = (V', E')$ cannot be distinguished by the GA-MLP with an operator family $\Omega$ that includes $\{D, AD^{-\alpha}\}$ under any choice of its parameters, it means that the two multisets $\{(d_i, \sum_{j \in \mathcal{N}(i)} d_j^{-\alpha}) : i \in V\}_m = \{(d_{i'}, \sum_{j' \in \mathcal{N}(i')} d_{j'}^{-\alpha}) : i' \in V'\}_m$, and therefore both of the following hold:

$$\{d_i : i \in V\}_m = \{d_{i'} : i' \in V'\}_m \tag{17}$$

$$\{\sum_{j \in \mathcal{N}(i)} d_j^{-\alpha} : i \in V\}_m = \{\sum_{j' \in \mathcal{N}(i')} d_{j'}^{-\alpha} : i' \in V'\}_m \tag{18}$$

To see what this means, we need the two following lemmas.

**Lemma 1.** *Let $\mathcal{S}_n$ be the set of all multisets consisting of at most $n$ elements, all of which are integers between $0$ and $n$. Consider the function $h_\alpha(S) := \sum_{u \in S} u^{-\alpha}$ defined for multisets $S$. If $\alpha > \frac{\log n}{\log n - \log(n-1)}$, $h_\alpha$ is an injective function on $\mathcal{S}_n$.*

*Proof of Lemma 1:* For $h_\alpha$ to be injective on $\mathcal{S}_n$, it suffices to require that $\forall l \leq n - 1$, there is $l^{-\alpha} > n(l+1)^{-\alpha}$, for which it is sufficient to require that $(n-1)^{-\alpha} > n^{-\alpha+1}$, or $\alpha > \frac{\log n}{\log n - \log(n-1)}$. □

**Lemma 2** (Babai et al. (1980), Theorem 1). *Consider the space of graphs with $n$ vertices, $\mathcal{G}_n$. There is a subset $\mathcal{K}_n \subseteq \mathcal{G}_n$ that contains almost all such graphs (i.e. the fraction converges to $1$ as $n \to \infty$) such that the following algorithm yields a unique identifier for every graph $G = (V, E) \in \mathcal{K}_n$:*

**Algorithm 1:** *Set $r = \lceil 3 \log n / \log 2 \rceil$, and let $\bar{d}(G)$ be the degree of the node in $V$ with the $r$th largest degree; For each node $i$ in $G$, define the multiset $\gamma_i = \{d_j : j \in \mathcal{N}(i), d_j > \bar{d}(G)\}_m$; Finally define a multiset associated with $G$, $F(G) = \{\gamma_i : i \in V\}_m$, which is the output of the algorithm.*

*In other words, $\forall G, G' \in \mathcal{K}_n$, $G$ and $G'$ are isomorphic if and only if $F(G) = F(G')$ as multisets. In particular, we can choose $\mathcal{K}_n$ such that the top $r$ node degrees of every graph in $\mathcal{K}_n$ are distinct.*

Based on these lemmas, we will show that when $\alpha > \frac{\log n}{\log n - \log(n-1)}$ and for $G, G' \in \mathcal{K}_n$, (17) and (18) together imply that $G$ is isomorphic to $G'$. To see this, suppose that (17) and (18) hold. Because of (17), we know that $G$ and $G'$ share the same degree sequence, and hence $\bar{d}(G) = \bar{d}(G')$. Because of (18), we know that there is a bijective map $\sigma$ from $V$ to $V'$ such that $\forall i \in V$,

$$\sum_{j \in \mathcal{N}(i)} d_j^{-\alpha} = \sum_{j' \in \mathcal{N}(i')} d_{j'}^{-\alpha} \,, \tag{19}$$

which, by Lemma 1, implies that $\{d_j : j \in \mathcal{N}(i)\}_m = \{d_{j'} : j' \in \mathcal{N}(i')\}_m$. We then have $\gamma_i = \{d_j : j \in \mathcal{N}(i)\}_m = \{d_j : j \in \mathcal{N}(i)\}_m \cap (\bar{d}(G), \infty) = \{d_{j'} : j' \in \mathcal{N}(i')\}_m \cap (\bar{d}(G'), \infty) = \gamma_{i'}$, and therefore $F(G) = F(G')$, which implies that $G$ and $G'$ are isomorphic by Lemma 2. This shows a contradiction. Therefore, if $G, G' \in \mathcal{K}_n$ are not isomorphic, then it cannot be the case that both (17) and (18) hold, and hence there exists a choice of parameters for the GA-MLP with $\{D, AD^{-\alpha}\} \subseteq \Omega$ that makes it return different outputs when applied to $G$ and $G'$. This proves Proposition 2. □

## G  PROOF OF PROPOSITION 3

As argued in the main text, to estimate the number of equivalence classes on $\mathcal{E}$ induced by GNNs, we need to estimate the number of possible rooted aggregation trees. In particular, to lower-bound the number of equivalence classes on $\mathcal{E}$ induced by GNNs, we only need to focus on a subset of all possible rooted aggregation trees, namely those in which every node has exactly $m$ children. Letting $\mathcal{T}_{m,K,\mathcal{X}}^{A}$ denote the set of all rooted aggregation trees of depth $K$ in which each non-leaf node has degree exactly $m$ and the node features belong to $\mathcal{X}$, we will first prove the following lemma:

**Lemma 3.** *If $|\mathcal{X}| \geq 2$, then $|\mathcal{T}_{m,K,\mathcal{X}}^{\mathrm{A}}| \geq (m-1)^{(2^K-1)}$.*

Note that a rooted *aggregation* tree needs to satisfy the constraint that each of its node must have its parent's feature equal to one of its children's feature, and so this lower bound is not as straightforward to prove as lower-bounding the total number of rooted subtrees. As argued above, this will allow us to derive Proposition 3.

*Proof of Lemma 3:* Define $\mathcal{B} := \{0, 1\}$. Since $|\mathcal{X}| \geq 2$, we assume without loss of generality that $\mathcal{B} \subseteq \mathcal{X}$. To prove a lower-bound on the cardinality of $\mathcal{T}_{m,K,\mathcal{X}}^{\mathrm{A}}$, it suffices to restrict our attention to its subset, $\mathcal{T}_{m,K}^{\mathrm{A}} := \mathcal{T}_{m,K,\mathcal{B}}^{\mathrm{A}}$, where all nodes have feature either 0 or 1. Furthermore, it is sufficient to restrict our attention to the subset of $\mathcal{T}_{m,K}^{\mathrm{A}}$ which contain all $2^K$ possible types of paths of length $K$ from the root to the leaves. Formally, with $\overline{\mathcal{W}}_k$ defined as in Appendix D, we let

$$\tilde{\mathcal{T}}_{m,K}^{\mathrm{A}} = \left\{ T \in \mathcal{T}_{m,K}^{\mathrm{A}} : \forall x_1, ..., x_K \in \mathcal{B}, \overline{\mathcal{W}}_K(T; (x_1, ..., x_K)) \geq 1 \right\}, \tag{20}$$

and it is sufficient to prove a lower bound on the cardinality of $\tilde{\mathcal{T}}_{m,K}^{\mathrm{A}}$. Define $\mathrm{P}_k = \{(x_1, ..., x_k) : x_1, ..., x_k \in \mathcal{B}\}$ to be the set of all binary $k$-tuples. By the definition of (20), we know that $\forall \tau \in \mathrm{P}_K, |\mathcal{W}_K(T; \tau)| \geq 1$. This means that $\forall \tau \in \mathrm{P}_K$, there exists at least one leaf node in $T$ such that the path from the root node to this node consists of a sequence of nodes with features exactly as given by $\tau$. We call any such node a *node under $\tau$*.

We show such a lower bound on the cardinality of $\tilde{\mathcal{T}}_{m,K}^{\mathrm{A}}$ inductively. For the base case, we know that $\tilde{\mathcal{T}}_{m,1}^{\mathrm{A}}$ consists of all binary-featured depth-1 rooted trees with at least 1 leaf node of feature 0 and 1 leaf node of feature 1, and hence $\tilde{\mathcal{T}}_{m,1}^{\mathrm{A}} = 2(m-1)$. Next, we consider the inductive step. For every $K \geq 1$ and every $T \in \tilde{\mathcal{T}}_{m,K}^{\mathrm{A}}$, we can generate rooted aggregation trees belonging to $T \in \tilde{\mathcal{T}}_{m,K+1}^{\mathrm{A}}$ by assigning children of feature 0 or 1 to the leaf nodes of $T$. First note that, from two non-isomorphic rooted aggregation trees $T$ and $T' \in \tilde{\mathcal{T}}_{m,K}^{\mathrm{A}}$, we obtain non-isomorphic rooted aggregation trees in $\tilde{\mathcal{T}}_{m,K+1}^{\mathrm{A}}$ in this way. Moreover, as we will show next, for every $T \in \tilde{\mathcal{T}}_{m,K}^{\mathrm{A}}$, we can lower-bound the number of distinct rooted aggregation trees belonging to $\tilde{\mathcal{T}}_{m,K+1}^{\mathrm{A}}$ obtained from $T$ in this way.

There are many choices to assign the children. To get a lower-bound on the cardinality of $\tilde{\mathcal{T}}_{m,K+1}^{\mathrm{A}}$, we only need to consider a subset of these choices of assignments, namely, those that assign the same number of children with feature 0 to every node under the same $\tau \in \mathrm{P}_K$. Thus, we let $\bar{q}_{K+1,\tau}$ denote the number of children of feature 0 assigned to every node in $\tau$. Due to the constraint that each node in the rooted aggregation tree must have its parent's feature equal to one of its children's feature, not all choices of $\{\bar{q}_{K+1,\tau}\}_{\tau \in \mathrm{P}_K}$ lead to legitimate rooted aggregation trees. Nonetheless, when restricting to the choices where $\forall \tau \in \mathrm{P}_K, 1 \leq \bar{q}_{K+1,\tau} \leq m-1$, we see that every leaf node of $T$ gets assigned at least one child of feature 0 and another child of feature 1, thereby satisfying the constraint above whether its parent has feature 0 or 1. Moreover, for such choices, the rooted aggregation tree of depth $K+1$ obtained in this way contains all $2^{K+1}$ possible paths of length $K+1$, and therefore belongs to $\tilde{\mathcal{T}}_{m,K+1}^{\mathrm{A}}$. Hence, it remains to show a lower bound on how many distinct trees in $\tilde{\mathcal{T}}_{m,K+1}^{\mathrm{A}}$ can be obtained in this way from each $T$. Since for $\tau, \tau' \in \mathrm{P}_K$ such that $\tau \neq \tau'$, a node under $\tau$ is distinguishable from a node under $\tau'$, we see that every legitimate choice of the tuple of $2^K$ integers, $(\bar{q}_{K+1,\tau})_{\tau \in \mathrm{P}_K}$, leads to a distinct rooted aggregation tree of depth $K+1$, and there are $(m-1)^{2^K}$ of these choices. Hence, we have derived that $|\tilde{\mathcal{T}}_{m,K+1}^{\mathrm{A}}| \geq (m-1)^{2^K} |\tilde{\mathcal{T}}_{m,K}^{\mathrm{A}}|$, and therefore $|\tilde{\mathcal{T}}_{m,K}^{\mathrm{A}}| \geq (m-1)^{\sum_{k=1}^{K} 2^k} = (m-1)^{2^K-1}$.

$\square$

# H PROOF OF PROPOSITION 4

According to the formula (3), by expanding the matrix product, we have

$$
\begin{aligned}
(\tilde{A}^k \varphi(X))_i &= \sum_{(i_1,...,i_k) \in \mathcal{W}_k(G^{[i]})} d_i^{-\alpha} d_{i_1}^{-(\alpha+\beta)}...d_{i_{k-1}}^{-(\alpha+\beta)} d_{i_k}^{-\beta} \varphi(X_{i_k}) \\
&= d_i^{-\alpha} \sum_{\substack{\{\bar{d}_1,...,\bar{d}_{t-1}\}_m, \\ \bar{d}_k, x}} \sum_{\substack{(i,i_1,...,i_k) \in \\ \mathcal{W}_k(G^{[i]};\{\bar{d}_1,...,\bar{d}_{k-1}\}_m, \bar{d}_k, x)}} (\bar{d}_1...\bar{d}_{k-1})^{-(\alpha+\beta)} \bar{d}_k \varphi(x) \\
&= d_i^{-\alpha} \sum_{\substack{\{\bar{d}_1,...,\bar{d}_{k-1}\}_m, \\ \bar{d}_k, x}} \left( (\bar{d}_1...\bar{d}_{k-1})^{-(\alpha+\beta)} \bar{d}_k \varphi(x) \right) \left| \mathcal{W}_k(G^{[i]};\{\bar{d}_1,...,\bar{d}_{t-1}\}_m, \bar{d}_k, x) \right| ,
\end{aligned}
\tag{21}
$$

with $\mathcal{W}_k(G^{[i]};\{\bar{d}_1,...,\bar{d}_{t-1}\}_m, \bar{d}_k, x)$ defined in Appendix D. Hence, for two different nodes $i$ in $G$ and $i'$ in $G'$ ($G$ and $G'$ can be the same graph), the node-wise outputs of the GA-MLP at $i$ and $i'$ will be identical if the rooted graphs $G^i$ and $G'^{[i']}$ satisfy $\mathcal{W}_k(G^{[i]};\{\bar{d}_1,...,\bar{d}_{k-1}\}_m, \bar{d}_k, x) = \mathcal{W}_k(G'^{[i']};\{\bar{d}_1,...,\bar{d}_{k-1}\}_m, \bar{d}_k, x)$ for every combination of choices on the multiset $\{\bar{d}_1,...,\bar{d}_{k-1}\}_m$, the integer $\bar{d}_k$ and the node feature $x$, under the constraints of $\bar{d}_1,...,\bar{d}_k \leq m$ and $x \in \mathcal{X}$. Note that there are at most $\binom{k+m-2}{m-1} \leq (k+m-2)^{m-1}$ possible choices of the multiset $\{\bar{d}_1,...,\bar{d}_{k-1}\}_m$, $m$ choices of $\bar{d}_k$ and $|\mathcal{X}|$ choices of $x$, thereby allowing at most $|\mathcal{X}|m(k+m-2)^{m-1}$ possible choices. Because of the constraint

$$
\sum_{\substack{\{\bar{d}_1,...,\bar{d}_{k-1}\}_m, \\ \bar{d}_k, x}} \left| \mathcal{W}_k(G_K^{[i]};\{\bar{d}_1,...,\bar{d}_{t-1}\}_m, \bar{d}_k, x) \right| = |\mathcal{W}_k(G^{[i]})| \leq m^k ,
\tag{22}
$$

We see that the total number of equivalence classes on $\mathcal{E}$ induced by such a GA-MLP is upper-bounded by $\binom{m^k + |\mathcal{X}|m(k+m-2)^{m-1}-1}{|\mathcal{X}|m(k+m-2)^{m-1}}$, which is on the order of $O(m^{k^m})$ with $k$ growing and $m$ bounded. Finally, since the total number of equivalence classes induced by multiple operators can be upper-bounded by the product of the number of equivalence classes induced by each operator separately, we derive the proposition as desired.

# I PROOF OF PROPOSITION 5

Consider the set of full $m$-ary rooted trees of depth $K$, $\mathcal{T}_{m,K,\mathcal{X}}$, that is all rooted trees of depth $K$ in which the nodes have features belonging to the discrete set $\mathcal{X} \subseteq \mathbb{N}$ and all non-leaf nodes have $m$ children. $\mathcal{T}_{m,K,\mathcal{X}}$ is a subset of $\mathcal{E}$, the space of all rooted graphs. If $f$ is a function represented by a GA-MLP using operators of at most $K$-hop, then for $T \in \mathcal{T}_{m,K,\mathcal{X}}$, we can write

$$
f(T) = \rho(\sum_{j \in V} a_j X_j) ,
\tag{23}
$$

where we denote the node set of $T$ by $V$ and the vectors $a_j$'s depend only on the topological relationship between $j$ and the root node. Let $V_k$ denote the set of nodes at depth $k$ of $T$. By the assumption that the operators depend only on the graph topology, and thanks to the topological symmetry of such full $m$-ary trees among all nodes on the same depth, we have that $\forall 1 \leq k \leq K$ and $\forall j, j' \in V_k$, there is $a_j = a'_j =: a_{[k]}$. Thus, we can write

$$
\begin{aligned}
f(T) &= \rho(\sum_{0 \leq k \leq K} \sum_{j \in V_k} a_{[k]} \phi(X_j)) \\
&= \rho(\sum_{0 \leq k \leq K} \sum_{x \in \mathcal{X}} \bar{a}_{[k],x} |\overline{\mathcal{W}}_k(T;x)|)
\end{aligned}
\tag{24}
$$

for some other set of coefficients $\bar{a}_{V_k,x}$'s, and where $\overline{\mathcal{W}}_k(T;x)$ is defined in Appendix D. In other words, for two trees $T$ and $T' \in \mathcal{T}_{m,K,\mathcal{X}}$, if $\forall 0 \leq k \leq K, \forall x \in \mathcal{X}$, they satisfy

$|\overline{\mathcal{W}}_k(T;x)| = |\overline{\mathcal{W}}_k(T';x)|$, then $f(T) = f(T')$ for all such $f$'s, and hence $T$ and $T'$ belong to the same equivalence class in $\mathcal{E}$ induced by GA-MLPs. Thus, for a certain subset of these equivalence classes, we can lower-bound the number of equivalence classes in $\mathcal{E}$ induced by GNNs that they intersect by lower-bounding the number of distinct trees in $\mathcal{T}_{m,K,\mathcal{X}}$ that they contain, because GNNs are able to distinguish non-isomorphic rooted subtrees. In particular, as a lower-bound is sufficient, we restrict attention to the subset of those trees with node features either 0 or 1, that is, trees belonging to $\mathcal{T}_{m,K} := \mathcal{T}_{m,K,\mathcal{B}}$, with $\mathcal{B} := \{0,1\}$.

In a rooted tree $T$, $\overline{\mathcal{W}}_k(T;x)$ gives the total number of nodes with feature $x$ at depth $k$. For integers $q_0, q_1, ..., q_K$ such that $0 \leq q_k \leq m^k, \forall k \leq K$, define

$$\mathcal{T}_{m,K,(q_0,q_1,...,q_K)} = \{T \in \mathcal{T}_{m,K} : \forall k \leq K, |\overline{\mathcal{W}}_k(T;0)| = q_k\} , \tag{25}$$

that is, the subset of trees whose *per-level-node-counts*, $\{|\overline{\mathcal{W}}_k(T;0)|\}_{k \leq K}$ (and therefore $\{|\overline{\mathcal{W}}_k(T;x)|\}_{k \leq K, x \in \mathcal{B}}$) are given by the tuple $(q_0, q_1, ..., q_k)$. From the argument above, all trees in the same $\mathcal{T}_{m,K,(q_0,q_1,...,q_K)}$ belong to the same equivalence class in $\mathcal{E}$ induced by GA-MLPs. On the other hand, every pair of non-isomorphic trees belong to different equivalence class in $\mathcal{E}$ induced by GNNs. Thus, to show Proposition 5, it is sufficient to find sufficiently many choices of $(q_0, q_1, ..., q_K)$ such that $\mathcal{T}_{m,K,(q_0,q_1,...,q_K)}$ contains sufficiently many non-isomorphic trees. Specifically, we will show the following:

**Lemma 4.** *For all integers $q_0, q_1, ..., q_K$ such that $\forall 2 \leq k \leq K$,*

$$2^k - 2^{k-2} \leq q_k \leq \frac{1}{2} m^k , \tag{26}$$

*there is*

$$|\mathcal{T}_{m,K,(q_0,q_1,...,q_K)}| \geq 2^{2^{K-1}-1} \tag{27}$$

*Proof of Lemma 4:* To prove such a lower bound on the cardinality of $\mathcal{T}_{m,K,(q_0,q_1,...,q_K)}$, it is sufficient to prove a lower bound on the cardinality of its subset,

$$\tilde{\mathcal{T}}_{m,K,(q_0,q_1,...,q_K)} = \{T \in \mathcal{T}_{m,K,(q_0,q_1,...,q_K)} : \forall x_1, ..., x_K \in \mathcal{B}, \overline{\mathcal{W}}_K(T;(x_1,...,x_K)) \geq 1\} . \tag{28}$$

A similar construction is involved in the proof of Lemma 3 in Appendix G. Then, we will prove this lemma by induction on $K$. For the base cases, it is obvious that $|\tilde{\mathcal{T}}_{m,0,(0)}| = |\tilde{\mathcal{T}}_{m,0,(1)}| = 1$, and $|\tilde{\mathcal{T}}_{m,1,(0,0)}| = |\tilde{\mathcal{T}}_{m,1,(0,1)}| = |\tilde{\mathcal{T}}_{m,1,(0,2)}| = |\tilde{\mathcal{T}}_{m,1,(1,0)}| = |\tilde{\mathcal{T}}_{m,1,(1,1)}| = |\tilde{\mathcal{T}}_{m,1,(1,2)}| = 1$. We next prove the inductive hypothesis that, for $K \geq 2$ and when $q_0, q_1, ..., q_K$ satisfying (26), there is

$$|\tilde{\mathcal{T}}_{m,K,(q_0,q_1,...,q_K)}| \geq 2^{2^{K-2}} \cdot |\tilde{\mathcal{T}}_{m,K-1,(q_0,q_1,...,q_{K-1})}| . \tag{29}$$

To see this, we will next show that $\forall T \in \tilde{\mathcal{T}}_{m,K-1,(q_0,q_1,...,q_{K-1})}$, we can generate enough number of depth-$K$ trees in $\tilde{\mathcal{T}}_{m,K,(q_0,q_1,...,q_K)}$ by appending children to the leaf nodes of $T$. Since any two depth-$K$ trees generated from two non-isomorphic depth-$(K-1)$ trees in this way are non-isomorphic, this will allow us to lower-bound the total number of trees in $\tilde{\mathcal{T}}_{m,K,(q_0,q_1,...,q_K)}$.

Consider the set of binary $k$-tuples, $\mathrm{P}_k = \{(x_1, ..., x_k) : x_1, ..., x_k \in \mathcal{B}\}$, of cardinality $2^k$. As $T \in \tilde{\mathcal{T}}_{m,K-1,(q_0,q_1,...,q_{K-1})}$, we know that $\forall \tau \in \mathrm{P}_{K-1}, |\overline{\mathcal{W}}_{K-1}(T;\tau)| \geq 1$. This means that $\forall \tau \in \mathrm{P}_k$, there exists at least one leaf node in $T$ such that the path from the root node to this node consists of a sequence of nodes with features given by $\tau$. We call any such node a *node under $\tau$*. The total number of the children of all nodes under $\tau$ is thus $m \cdot |\overline{\mathcal{W}}_{K-1}(T;\tau)| \geq m$. Thus, the total number of children with feature 0 of all nodes under $\tau$ is bounded between 0 and $m \cdot |\overline{\mathcal{W}}_{K-1}(T;\tau)|$. Conversely, for any $2^{K-1}$-tuple of non-negative integers, $(\bar{q}_{K,\tau})_{\tau \in \mathrm{P}_{K-1}}$, which satisfy $\forall \tau \in \mathrm{P}_{K-1}, 1 \leq \bar{q}_{K,\tau} \leq m \cdot |\overline{\mathcal{W}}_{K-1}(T;\tau)| - 1$ can be "realized" by at least some depth-$K$ tree $T'$ obtained by appending children to the leaf nodes of $T$, in the sense that $\forall \tau = (x_1, ..., x_{K-1}) \in \mathrm{P}_{K-1}$, there is $\overline{\mathcal{W}}_K(T';(x_1,...,x_{K-1},0)) = \bar{q}_{K,\tau}$ and $\overline{\mathcal{W}}_K(T';(x_1,...,x_{K-1},1)) = m \cdot |\overline{\mathcal{W}}_{K-1}(T;\tau)| - \bar{q}_{K,\tau}$, and hence $T' \in \mathcal{T}_{m,K,(q_0,...,q_{K-1},\bar{q}_K)}$, with $\bar{q}_K = \sum_{\tau \in \mathrm{P}_{K-1}} \bar{q}_{K,\tau}$. Because of the requirement that $1 \leq \bar{q}_{K,\tau} \leq m \cdot |\overline{\mathcal{W}}_{K-1}(T;\tau)| - 1$, we further have that $\forall \tau' \in \mathrm{P}_K, \overline{\mathcal{W}}_K(T',\tau') \geq 1$, which implies that $T' \in \tilde{\mathcal{T}}_{m,K,(q_0,...,q_{K-1},\bar{q}_K)}$. Therefore, for some fixed $q_K$, in order to lower-bound the cardinality of $\tilde{\mathcal{T}}_{m,K,(q_0,...,q_K)}$ by that of $\tilde{\mathcal{T}}_{m,K-1,(q_0,...,q_{K-1})}$, it is sufficient to show a lower bound (which

is uniform for all $T \in \tilde{\mathcal{T}}_{m,K-1,(q_0,q_1,\ldots,q_{K-1})}$) on the number of $2^{K-1}$-tuples, $(q_{K,\tau})_{\tau \in \mathrm{P}_{K-1}}$, which satisfy

$$q_K = \sum_{\tau \in \mathrm{P}_{K-1}} q_{K,\tau} \tag{30}$$

$$\forall \tau \in \mathrm{P}_{K-1}, 1 \leq q_{K,\tau} \leq m \cdot |\overline{\mathcal{W}}_{K-1}(T;\tau)| - 1$$

A simple bound can be obtained in the following way. For every such $T$, we sort the $2^{K-1}$-tuples in $\mathrm{P}_{K-1}$ in ascending order of $|\overline{\mathcal{W}}_{K-1}(T;\cdot)|$, and define $\mathrm{P}'_{K-1,T}$ to be the subset of the first $2^{K-2}$ of these elements according to this order. Thus, for example, $\forall \tau \in \mathrm{P}'_{K-1,T}, \forall \tau' \in \mathrm{P}_{K-1} \setminus \mathrm{P}'_{K-1,T}$, there is $|\overline{\mathcal{W}}_{K-1}(T;\tau)| \leq |\overline{\mathcal{W}}_{K-1}(T;\tau')|$. As a consequence, we have $\sum_{\tau \in \mathrm{P}'_{K-1,T}} |\overline{\mathcal{W}}_{K-1}(T;\tau)| \leq \sum_{\tau \in \mathrm{P}_{K-1} \setminus \mathrm{P}'_{K-1,T}} |\overline{\mathcal{W}}_{K-1}(T;\tau)|$, and so $\sum_{\tau \in \mathrm{P}'_{K-1,T}} |\overline{\mathcal{W}}_{K-1}(T;\tau)| \leq \frac{1}{2} \sum_{\tau \in \mathrm{P}_{K-1}} |\overline{\mathcal{W}}_{K-1}(T;\tau)| = \frac{1}{2} m^{K-1} \leq \sum_{\tau \in \mathrm{P}_{K-1} \setminus \mathrm{P}'_{K-1,T}} |\overline{\mathcal{W}}_{K-1}(T;\tau)|$.

**Lemma 5.** *Let $K \geq 2$ and $q_K$ satisfy (26). Then for all choices of the $2^{K-2}$-tuple of integers, $(\bar{q}_{K,\tau})_{\tau \in \mathrm{P}'_{K-1,T}}$, such that $\forall \tau \in \mathrm{P}'_{K-1,T}, \bar{q}_{K,\tau} = 1$ or 2, we can complete it into at least one $2^{K-1}$-tuple of integers, $(\bar{q}_{K,\tau})_{\tau \in \mathrm{P}_{K-1}}$, which satisfy (30).*

*Proof of Lemma 5:* For any such $2^{K-2}$-tuple, $(\bar{q}_{K,\tau})_{\tau \in \mathrm{P}'_{K-1,T}}$, in order to satisfy the constraints of (30), it is sufficient to find another $2^{K-2}$ integers, $(\bar{q}_{K,\tau})_{\tau \in \mathrm{P}_{K-1} \setminus \mathrm{P}'_{K-1,T}}$, which satisfy

$$\sum_{\tau \in \mathrm{P}_{K-1} \setminus \mathrm{P}'_{K-1,T}} \bar{q}_{K,\tau} = q_K - \sum_{\tau \in \mathrm{P}'_{K-1,T}} \bar{q}_{K,\tau} \tag{31}$$

$$\forall \tau \in \mathrm{P}_{K-1} \setminus \mathrm{P}'_{K-1,T}, 1 \leq \bar{q}_{K,\tau} \leq m \cdot |\overline{\mathcal{W}}_{K-1}(T;\tau)| - 1 \tag{32}$$

On one hand, since $\bar{q}_{K,\tau} = 1$ or 2, $\forall \tau \in \mathrm{P}'_{K-1,T}$, there is $q_K - 2^{K-1} \leq q_K - \sum_{\tau \in \mathrm{P}'_{K-1,T}} \bar{q}_{K,\tau} \leq q_K - 2^{K-2}$. On the other hand, with the only other constraint being (32), it is possible to find $(\bar{q}_{K,\tau})_{\tau \in \mathrm{P}_{K-1} \setminus \mathrm{P}'_{K-1,T}}$ such that $\sum_{\tau \in \mathrm{P}_{K-1} \setminus \mathrm{P}'_{K-1,T}}$ equals any integer between $2^{K-2}$ and $m \cdot \sum_{\tau \in \mathrm{P}_{K-1} \setminus \mathrm{P}'_{K-1,T}} |\overline{\mathcal{W}}_{K-1}(T;\tau)| - 2^{K-2}$, and hence any integer between $2^{K-2}$ and $m \cdot \frac{1}{2} m^{K-1} - 2^{K-2} = \frac{1}{2} m^K - 2^{K-2}$. Hence, as long as $2^K - 2^{K-2} \leq q_K \leq \frac{1}{2} m^K$, which is the assumption of (26), Lemma 5 holds true. □

Lemma 5 implies that $\forall T \in \tilde{\mathcal{T}}_{m,K-1,(q_0,q_1,\ldots,q_{K-1})}$, there are at least $2^{2^{K-2}}$ distinct choices of $2^{K-1}$-tuples $(q_{K,\tau})_{\tau \in \mathrm{P}_{K-1}}$ that satisfy the constraint of (30), and hence at least $2^{2^{K-2}}$ non-isomorphic trees in $\tilde{\mathcal{T}}_{m,K,(q_0,q_1,\ldots,q_{K-1},q_K)}$ obtained by appending children to the leaf nodes of $T$. This proves the inductive hypothesis. Hence, we have

$$|\tilde{\mathcal{T}}_{m,K,(q_0,q_1,\ldots,q_K)}| \geq \prod_{k=2}^{K} 2^{2^{k-2}} = 2^{2^{K-1}-1}, \tag{33}$$

which implies Lemma 4. □

Since $m \geq 2$ by assumption, $\frac{1}{2} m^K - (2^K - 2^{K-2})$ grows exponentially in $K$. This proves Proposition 5.

## J  PROOF OF PROPOSITION 6

Since the number of walks of a particular type that has length at most $k$ is completely determined by the rooted aggregation tree structure of depth $k$, it is straightforward to see that all egonets in the same equivalence class induced by $k$ iterations of WL (and therefore GNNs of depth $k$), which yield the same rooted aggregation tree, will get mapped to the same value by $f_k$.

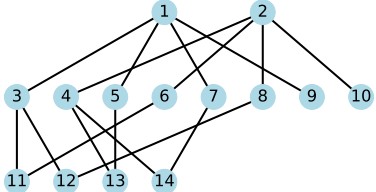 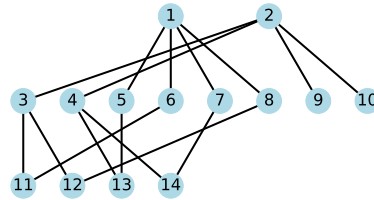

Figure 5: A pair of graphs with identical node features, $G$ (left) and $G'$ (right), which can be distinguished by 2 iterations of the WL test but not by the GA-MLP with $\Omega \subseteq \{A^k\}_{k \in \mathbb{N}}$.

For the second part of the claim pertaining to GA-MLPs, we assume for simplicity that $\mathcal{X} = \mathcal{B} = \{0, 1\}$, as the extension to the general case is straightforward but demanding heavier notations. Following the strategy in the proof of Proposition 5, it is sufficient to find exponentially-in-$k$ many choices of the tuple $(q_0, q_1, ..., q_k)$, with $0 \leq q_k \leq m^k$, such that the image of $\mathcal{T}_{m,k,(q_0,q_1,...,q_k)}$ (as defined in (25)) under $f_k$ contains exponentially-in-$k$ many values.

To make it simpler to refer to different nodes in the tree, we index each node in a rooted tree by a tuple of natural numbers: for example, the index-tuple $[1, 3, 2]$ refers to the node at depth $3$ that is the second children of the third children of the first children of the root. Since there is no intrinsic ordering to different children of the same node, there exist multiple ways of consistently indexing the nodes in a rooted tree. However, to specify a tree, it suffices to specify the node features of all nodes under *one* such way of indexing.

Given $x_1, ..., x_k \in \mathcal{B}$, we consider a set of depth-$k$ full $m$-ary trees that satisfy the following: $\forall k' \leq k - 1$ and $l_1, ..., l_{k'} \in [m]$, $x_{[l_1, l_2, ..., l_{k'}]} = x_{k'}$ if $l_1 = 1$ and $\neg x_{k'}$ if $l_1 > 1$. Note that these trees satisfy, for $k' \leq k - 1$, $q_{k'} = m^{k'-1}$ if $x_{k'} = 0$ and $q_{k'} = (m-1)m^{k'-1}$ if $x_{k'} = 1$. Thus, $\forall l_2, ..., l_k \in [m]$, the node $[1, l_2, ..., l_k]$ is under the path $\tau = (x_1, ..., x_k)$ if and only if $x_{[1, l_2, ..., l_k]} = x_k$, whereas for $l_1 > 1$, the node $[l_1, l_2, ..., l_k]$ is not under the path $\tau$ regardless of the feature of $[l_1, l_2, ..., l_k]$. Therefore, $f_k(G^{[i]}) = |\mathcal{W}_k(G^{[i]}; (x_1, ..., x_k))|$ equals the number of node of feature $x_k$ among the set of $m^{k-1}$ nodes, $\{[1, l_2, ..., l_k]\}_{l_2, ..., l_k \in [m]}$. Hence, if for $k' \leq k - 1$, we set $q_{k'} = m^{k'-1}$ if $x_{k'} = 0$ and $q_{k'} = (m-1)m^{k'-1}$ if $x_{k'} = 1$, then choosing any $q_k$ between $m^{k-1}$ and $(m-1)m^{k-1}$, we have that for every integer between $0$ and $m^{k-1}$, there exists a tree $T$ in $\mathcal{T}_{m,k,(q_0,...,q_k)}$ such that $f_k(T)$ equals this integer. Since there are $(m-2)m^{k-1}$ choices of $q_k$ (and therefore the tuple $(q_0, ..., q_k)$) and $m^{k-1} + 1$ values in the image of $\mathcal{T}_{m,k,(q_0,...,q_k)}$ under $f_k$, this proves the proposition.

## K  PROOF OF PROPOSITION 1

We will first prove that the pair of graphs cannot be distinguished by any GA-MLP with $\Omega \subseteq \{A^k\}_{k \in \mathbb{N}}$. Let $X$ and $A$, $X'$ and $A'$ be the node feature vector and adjacency matrix of the two graphs, $G$ and $G'$, respectively. As these two graphs both contain $14$ nodes that have identical features, we have $X, X' \in \mathbb{R}^{14 \times 1}$ both being the all-1 vector. Moreover, $\forall i \in [14]$,

$$(A^k X)_i = w_k(i), \quad ((A')^k(X'))_i = w'_k(i) \tag{34}$$

where we use $w_k(i)$ and $w'_k(i)$ to denote the numbers of walks (allowing backtracking) of length $k$ starting from node $i$ in graphs $G$ and $G'$, respectively. Thus, to show that any GA-MLP with $\Omega \subseteq \{A^k\}_{k \in \mathbb{N}}$ necessarily returns the same output on $G$ and $G'$, it is sufficient to show that $\forall k \in \mathbb{N}$, $A^k X = (A')^k (X')$, and therefore sufficient to show that $\forall k \in \mathbb{N}$ and $\forall i \in [14]$, there is $w_k(i) = w'_k(i)$. In fact, we will prove the following lemma:

**Lemma 6.** $\forall k \in \mathbb{N}$,

$$w_k(i) = w'_k(i), \ \forall i \in [14] \tag{35}$$
$$w_k(1) = w_k(2) \tag{36}$$
$$w_k(3) + w_k(9) = w_k(6) + w_k(8) \tag{37}$$
$$w_k(5) + w_k(7) = w_k(4) + w_k(10) \tag{38}$$

*Proof of Lemma 6:* We prove this lemma by induction. For the base case, we have that $w_0(i) = w'_0(i), \forall i \in [14]$. Next, we assume that (35) - (38) hold for some $k \in \mathbb{N}$ and prove it for $k + 1$. A first property to note is that $\forall k \in \mathbb{N}$, $w_{k+1}(i) = \sum_{j \in \mathcal{N}(i)} w_k(j)$ and $w'_{k+1}(i) = \sum_{j \in \mathcal{N}'(i)} w'_k(j)$, where we use $\mathcal{N}(i)$ and $\mathcal{N}'(i)$ to denote the neighborhood of $i$ in $G$ and $G'$, respectively.

To show (35) for $k + 1$, we look at each node separately:

- $i = 1$

$$\begin{aligned} w_{k+1}(1) =& w_k(3) + w_k(5) + w_k(7) + w_k(9) \\ =& w_k(5) + w_k(6) + w_k(7) + w_k(8) \\ =& w'_k(5) + w'_k(6) + w'_k(7) + w'_k(8) \\ =& w'_{k+1}(1) \end{aligned} \tag{39}$$

- $i = 2$

$$\begin{aligned} w_{k+1}(2) =& w_k(4) + w_k(6) + w_k(8) + w_k(10) \\ =& w_k(3) + w_k(4) + w_k(9) + w_k(10) \\ =& w'_k(3) + w'_k(4) + w'_k(9) + w'_k(10) \\ =& w'_{k+1}(2) \end{aligned} \tag{40}$$

- $i = 3$

$$\begin{aligned} w_{k+1}(3) =& w_k(1) + w_k(11) + w_k(12) \\ =& w_k(2) + w_k(11) + w_k(12) \\ =& w'_k(2) + w'_k(11) + w'_k(12) \\ =& w'_{k+1}(3) \end{aligned} \tag{41}$$

- $i = 4$

$$\begin{aligned} w_{k+1}(4) =& w_k(2) + w_k(13) + w_k(14) \\ =& w'_k(2) + w'_k(13) + w'_k(14) \\ =& w'_{k+1}(4) \end{aligned} \tag{42}$$

- $i = 5$

$$\begin{aligned} w_{k+1}(5) =& w_k(1) + w_k(13) \\ =& w'_k(1) + w'_k(13) \\ =& w'_{k+1}(5) \end{aligned} \tag{43}$$

- $i = 6$

$$\begin{aligned} w_{k+1}(6) =& w_k(2) + w_k(11) \\ =& w_k(1) + w_k(11) \\ =& w'_k(1) + w'_k(11) \\ =& w'_{k+1}(6) \end{aligned} \tag{44}$$

- $i = 7$

$$\begin{aligned} w_{k+1}(7) =& w_k(1) + w_k(13) \\ =& w'_k(1) + w'_k(13) \\ =& w'_{k+1}(7) \end{aligned} \tag{45}$$

- $i = 8$

$$
\begin{aligned}
w_{k+1}(8) =& w_k(2) + w_k(12) \\
=& w_k(1) + w_k(12) \\
=& w'_k(1) + w'_k(12) \\
=& w'_{k+1}(8)
\end{aligned}
\tag{46}
$$

- $i = 9$

$$
\begin{aligned}
w_{k+1}(9) =& w_k(1) \\
=& w_k(2) \\
=& w'_k(2) \\
=& w'_{k+1}(9)
\end{aligned}
\tag{47}
$$

- $i = 10$

$$
\begin{aligned}
w_{k+1}(10) =& w_k(2) \\
=& w'_k(2) \\
=& w'_{k+1}(10)
\end{aligned}
\tag{48}
$$

- $i \in \{11, ..., 14\}$
  For each of these $i$'s, $\mathcal{N}(i) = \mathcal{N}'(i)$. Therefore,

$$
\begin{aligned}
w_{k+1}(i) =& \sum_{j \in \mathcal{N}(i)} w_k(j) \\
=& \sum_{j \in \mathcal{N}'(i)} w'_k(j) \\
=& w'_{k+1}(i)
\end{aligned}
\tag{49}
$$

Next, for (36) - (38) at $k + 1$,

$$
\begin{aligned}
w_{k+1}(1) =& w_k(3) + w_k(5) + w_k(7) + w_k(9) \\
=& w_k(4) + w_k(6) + w_k(8) + w_k(10) \\
=& w_{k+1}(2)
\end{aligned}
\tag{50}
$$

$$
\begin{aligned}
w_{k+1}(3) + w_{k+1}(9) =& 2w_k(1) + w_k(11) + w_k(12) \\
=& 2w_k(2) + w_k(11) + w_k(12) \\
=& w_{k+1}(6) + w_{k+1}(8)
\end{aligned}
\tag{51}
$$

$$
\begin{aligned}
w_{k+1}(5) + w_{k+1}(7) =& 2w_k(1) + w_k(13) + w_k(14) \\
=& 2w_k(2) + w_k(13) + w_k(14) \\
=& w_{k+1}(4) + w_{k+1}(10)
\end{aligned}
\tag{52}
$$

This proves the inductive hypothethis for $k + 1$. $\qquad\square$

We next argue that these two graphs can be distinguished by WL in 2 iterations. This is because 2 iterations of WL distinguish neighborhoods up to the depth-2 rooted aggregation trees (as will be defined in Section 5), and it is not hard to see that the multiset of depth-2 rooted aggregation trees are different for the two graphs. Note that a depth-2 rooted subtree can be represented by the multiset of the degrees of the depth-1 children. Then for example, the depth-2 rooted aggregation trees of 1 and 2 in $G$ are both $\{3, 2, 2, 1\}_m$, while their rooted aggregation trees in $G'$ are $\{2, 2, 2, 2\}_m$ and $\{3, 3, 1, 1\}_m$, respectively.

## L  EXPERIMENT DETAILS

### L.1  SPECIFIC ARCHITECTURES

In Section 6, we show experiments on several tasks to confirm our theoretical results with several related architectures. Here are some explanations for them:

- **GIN**: Graph Isomorphism Networks proposed by Xu et al. (2019). In our experiment of counting attributed walks, we take the depth of GIN as same as the depth of target walks. The number of hidden dimensions is searched in $\{8, 16, 32, 64, 256\}$. The model is trained with the Adam optimizer (Kingma & Ba, 2014) with learning rate selected from $\{0.1, 0.02, 0.01, 0.005, 0.001\}$. We also train a variant with Jumping Knowledge (Xu et al., 2018).

- **sGNN**: Spectral GNN proposed by Chen et al. (2019b), which can be viewed as a learnable generalization of power iterations on a collection of operators. While the best performing variant utilizes the non-backtracking operator on the line graph, for a fairer comparison with GA-MLPs, we choose a variant with the base collection of operators being $\{I, A, A^2\}$ on each layer and depth 60, which then has the same receptive field as the chosen GA-MLP models. The model is trained with the Adam optimizer with learning rate selected from $\{0.001, 0.002, 0.004\}$.

- **GA-MLP**: a multilayer perceptron following graph augmented features. For counting attributed walks, we choose the operators from $\{I, \bar{A}_{\{\epsilon\}}^k\}$. The number of hidden dimensions is searched in $\{8, 32, 64, 256\}$. We take the highest order of operators as the twice depth of target walks at most. For comminity detection, we choose the operators from $\{I, \bar{A}_{\{\epsilon\}}^k, \tilde{H}^k\}$ where $\tilde{H}$ is induced from the Bethe Hessian matrix $H$. The highest order of operators is searched in $\{30, 60, 120\}$. The number of hidden dimensions is searched in $\{10, 20\}$. On both tasks, the model is trained with the Adam optimizer (Kingma & Ba, 2014) with learning rate selected from $\{0.1, 0.02, 0.01, 0.005, 0.001, 0.0001\}$. Additionally, we use Batch Normalization (Ioffe & Szegedy, 2015) in community detection after propagating through each operator, following the normalization strategy from Chen et al. (2019b). We choose $\varphi$ to be the identity function.

## L.2 BETHE HESSIAN

The Bethe Hessian matrix is defined as

$$H(r) := (r^2 - 1)I - rA + D.$$

with $r$ being a flexible parameter. In SBM, an optimal choice is $r_c = \sqrt{c}$, where $c$ is the average degree. Spectral clustering (Saade et al., 2014) can be performed by computing the eigenvectors associated with the negative eigenvalues of $H(r_c)$ to get clustering information in assortative binary stochastic block model, which is the scenario we consider. In order to utilize power iterations for eigenvector extraction, we induce a new matrix $\tilde{H}$ as

$$\tilde{H} := \kappa I - H(r_c),$$

so that the smallest eigenvalues of $H$ become the largest eigenvalues of $\tilde{H}$. We choose $\kappa = 8$ in our experiments. For GA-MLP-$H$, we then let $\Omega = \{I, \tilde{H}, ..., \tilde{H}^K\}$.

## L.3 RESULTS FOR GA-MLP-$\tilde{A}_{(1)}$ IN COMMUNITY DETECTION

Table 4: Results for community detection on binary SBM by GA-MLP-$\tilde{A}_{(1)}$

| Rank of hardness | 1 | 2 | 3 | 4 | 5 |
|---|---|---|---|---|---|
| Overlap | 0.128 | 0.164 | 0.262 | 0.707 | 0.563 |

