# OpenReview forum: "On Graph Neural Networks versus Graph-Augmented MLPs"
_ICLR.cc/2021/Conference — ICLR 2021 Poster_

### Official Review · AnonReviewer4 · 2020-10-28
**Very insightful theoretical analysis, it'd be helpful to attempt to interpret for more practical problems**

**Rating:** 7
**Confidence:** 3

**Review:**

The paper presents a theoretical analysis to compare expressive power of Graph-Neural Networks (GNNs) w.r.t a class of simpler graph modles called  Graph-Augmented MLPs (GA-MLPs).   GNNs, especially deeper ones can be more difficult to train, and GA-MLPs have a simpler structure, significantly easier to train, and have been shown to have competitive performance on a number of tasks.  The paper dives deep into several problems (graph-isomorphism, node-classification, and community detection) and through innovative analysis shows that GNNs at least theoretically can have significant advantages for some of the problems.

I enjoyed reading this paper, and in my opinion it's an important contribution to the field.  It's much more common to see papers that do only experimental validations  (which are of course very important in their own right), but little attempt is usually done beyond simple intuition.  This paper employs a number of technical tools to formally establish that for some problems GNNs (at least in theory, and conditioned on being able to learn them well) are significantly more expressive than GA-MLPs.  Some of the discussion can be made clearer,  but overall the paper is very well written.  Experimental validation is fairly basic, but helpful, however, it's mostly on stylized problems, and a general outlook for practical graph-ML problems would be helpful.

Some comments:
1)  The applications you consider (apart from community detection) seem quite stylized, (e.g. counting the number of certain types of walks from a node), so it would be nice to make some comments on what your analysis may imply  for typical practical applications of GNNs  (e.g. combinatorial optimization / graph or node classification / various tasks of using graph NNs in material science / chemistry  and give some comments of where do you feel GA-MLPs may be competitive.
2) You also mentioned that there's recent progress in training deeper GNNs (using residual connections, and other tricks), so the outlook from the paper seems to be that for now GNNs are harder to tune, but more expressive, but the hope is that with advances in training that won't be an issue,  and then GA-MLPs will be less competitive, except on certain applications like graph-isomorphism?
3)  In section 4,  it seems that distinguishing almost any pair of graphs drawn from a uniform distribution is not a very complex tasks -- as such pairs of graphs are likely to be very different.  Would it be harder to distinguish two graphs, where the second graph starts from the first and includes some modifications / transformations (e.g. swapping edges, e.t.c).
4) last sentence before the summary in section 1 seems to have grammatical issues.
5) It could be useful to give a short motivating sentence for Bethe-Hessian (e.g. is connected to Bethe-free-energy?)
6) What is laplacian sub-eigenspaces -- do you mean laplacian eigen sub-spaces?  (section  2).
7)  I liked the analysis based on attributed walks , and rooted aggregated trees,  it may have some connections to computation-trees and walk-based type of analysis in graphical models literature:
e.g. self-avoding walks for graphs and independent sets,  Chandrasekaran, Chertkov, Gamarnik et al, "Counitng independent sets using the Bethe Approximation",  and "Walk-Sums and Belief Propagation in Gaussian Graphical Models", johnson, malioutov, willsky, JMLR, 2006,  and  Yedidia, Freeman, Weiss, "understanding belief propagation and its generalizations".

---

> ### Author Response · Authors · 2020-11-24
> **Authors' response**
>
> Thanks for your kind comments and various interesting suggestions!
>
> 1. _On “what your analysis may imply for typical practical applications of GNNs”_:
>
> Indeed, we hope to shed some light on the types of practical tasks on which GNNs have an edge over GA-MLPs. As we showed both in theory and in experiments, GA-MLPs are not expressive enough for counting attributed walks in graphs, and hence applications where this task is explicitly or implicitly involved could show a gap between GNNs and GA-MLPs. For example, in the context of knowledge graph reasoning, it has been shown in [1] that GNNs are able to encode any logical rule corresponding to a path (which is an attributed walk in our language). Based on our theory, these logical rules cannot be exactly encoded by any GA-MLP model adapted to such a task. Another example in the context of learning combinatorial algorithms is complex dynamical programming algorithms, with which GNNs are known to have good algorithmic alignment ([2]). More generally, the counting of graph substructures has been known to be relevant to graph-based tasks in molecular predictions ([3]), computational biology ([4]) as well as social network analysis ([5]), and we expect to see a gap between the performances of GNNs and GA-MLPs on many of these tasks as well. In addition, community detection is still an important problem in social network analysis, and there have been recent interests in incorporating node feature information into the task, for which GNNs can be very helpful ([6, 7]).
>
>
> 2. _On the implications of “recent progress in training deeper GNNs”_:
>
> With the series of interesting progress in this direction, the usage of deep GNNs becomes more realistic, which makes a theoretical study on the role of depth in their expressive powers more relevant. Meanwhile, though the training quality can be improved, we still expect some other disadvantages of using deeper GNNs, such as high computational complexity. In comparison, GA-MLP models can take advantage of pre-computations of the feature augmentation stage, which makes them a lot more scalable to larger graphs. Therefore, an understanding of the gap in expressive power between deep GNNs and GA-MLPs is crucial: if a task lies in this gap, then we are likely to see an advantage of GNNs over GA-MLPs on it; otherwise, GA-MLPs could be preferred over deep GNNs from the viewpoint of scalability.
>
> 3. _On whether it is interesting to see whether models can distinguish graphs that are modifications of each other instead of independently sampled graphs_:
>
> Yes, this sounds like an interesting idea for assessing the expressive power of GNN models, since precisely as we were arguing in the paper, distinguishing random graph pairs does not seem that hard. We are not aware of works in this direction in the graph isomorphism literature, and it can be quite worthy of exploring.
>
> 4. _On Bethe Hessian and and Bethe free energy_:
>
> Indeed, the Bethe Hessian matrix is the Hessian matrix of the Bethe free energy, and the appearance of a negative eigenvalue of the Bethe Hessian corresponds to the appearance of a cluster in the graph [8].
>
> 5. _On the walk-based analysis in graphical models literature_:
>
> Thanks for pointing us to these interesting results! We should first admit that they are beyond our current expertise, but it feels tempted to interpret them as indicating that the counting of attributed walks could be relevant to tasks like counting independent sets or graphical model inferences. Does that seem like the correct interpretation? We are quite curious to hear if you have any further insights to share on this.
>
> Last but not least, we appreciate that you pointed out those typos to us.
>
> References:
>
> [1] Teru, Denis, Hamilton (2020). "Inductive Relation Prediction by Subgraph Reasoning". _ICML 2020_.
>
> [2] Xu, Li, Zhang, Du, Kawarabayashi, Jegelka (2020). "What Can Neural Networks Reason About?" _ICLR 2020_.
>
> [3] Murray and Rees (2009). "The rise of fragment-based drug discovery". _Nature Chemistry_.
>
> [4] Koyutürk, Grama, Szpankowski (2004). "An efficient algorithm for detecting frequent subgraphs in biological networks." _Bioinformatics_.
>
> [5] Jiang, Coenen, Zito (2010). "Finding frequent subgraphs in longitudinal social network data using a weighted graph mining approach." _International Conference on Advanced Data Mining and Applications_.
>
> [6] Chunaev (2020). "Community detection in node-attributed social networks: a survey". _Computer Science Review_.
>
> [7] Chen, Li, Bruna (2019). "Supervised community detection with line graph neural networks". _ICLR 2019_.
>
> [8] Saade, Krzakala, Zdeborova (2014). "Spectral Clustering of Graphs with the Bethe Hessian". _NIPS 2014_.

---

### Official Review · AnonReviewer1 · 2020-10-28
**A very good paper about expressive power of GNN vs GA-GNN.**

**Rating:** 8
**Confidence:** 3

**Review:**

The paper is a theoretical analysis of two different classes of graph neural network: 1) GNN based on neighborhood aggregations (GNN) and 2) feature augmentation before MLP (GA-MLP).
The author contributions may be summarized as:
- exhibiting graphs/problems that may be handled by one method and not the other,
- giving an upper-bound on the number of equivalent clases induceds by linear GA-MLP in term of walks in a rooted tree.
- showing a gap in expressivness between GNN and GA-MLP
- showing that the choice of operator in GA-MLP is crucial and may have expressive power beyond WL.

Pros:
+ The paper is well written.
+ The theoretical analysis performed in the paper is new, non trivial, and very interesting, giving light on these two architectures.

Remarks:
- Section 3.1: shouldn't 2-EXP be: log log is polynomial (instead of linear)?
- Section 5, corollary 1: 'poly-exponential'?

Questions:
- How to go beyond linear operators in section 5?

Typos:
- after proposition 6: in words -> on other words
- few sentences without verb and missing punctuation in the appendix
- many references are incomplete, e.g. On the universality of invariant networks is ICML'19, What graph neural networks cannot learn: depth vs width is ICLR'20, etc

---

> ### Author Response · Authors · 2020-11-24
> **Authors' response**
>
> Thank you very much for the comments and suggestions! We’re glad you found our work promising.
>
> The question of going beyond linear operators is a great one. For two of the results in Section 5 - the first half of Proposition 5 (that there exists an exponential-in-depth gap between the equivalence classes of rooted graphs induced by GNN and the GA-MLP) and Proposition 6 (that the GA-MLP cannot exactly count attributed walks) - we can indeed extend from equivariant linear operators to **arbitrary** equivariant (not necessarily linear) operators that are intrinsic to the graph topology, and we prove this in Appendix A in the revised version of the paper. The intuition is the following: the proofs are built upon examples of rooted graphs with a high degree of symmetry in its topology, meaning that there are many sets of nodes such that exchanging the features among the nodes in these sets will not affect the output of a GA-MLP on this rooted graph, if the GA-MLP uses operators that are intrinsic to the graph topology (whether linear or nonlinear). Hence, heuristically, the GA-MLP loses much more information compared to an GNN, which can be captured by the number of equivalence classes on the rooted graphs.
>
> Also, thanks for catching the typos!

---

### Official Review · AnonReviewer3 · 2020-10-29
**GNNs vs Graph-Augmented MLPs**

**Rating:** 5
**Confidence:** 4

**Review:**

The paper studies a variant of Graph Neural Networks (GNNs) namely, Graph Augmented MLPs (GA-MLPs). Unlike in GNNs where nodes send messages to neighbors, and aggregate received messages via non-linear MLPs,  GA-MLPs rely on a single augmented embedding computed once and then applying an MLP to the new embeddings. The augmented embeddings can be obtained by applying linear transformations of the form A, A^2, …, A^k to the input representations, thereby capturing larger neighborhoods. The main goal of the paper is to demonstrate a fundamental weakness when using GA-MLPs for solving graph problems as compared to GNNs. Along these line the paper the main results can be characterized as follows:
       1) The paper identifies a specific instance of identifying non-isomorphic graphs that can be solved via a GNN but not by a GA-MLP framework.
       2) The paper provides an empirical and experimental evaluation of the representation power of GNNs versus Graph-Augmented MLPs, and show a separation in expressive power between the two in terms of node level functions on rooted graphs. Specifically, they show that the set of functions that can be represented by a GNN of a certain depth) grows doubly exponentially in k, as opposed to only exponential growth of the function class when considering a similar GA-MLP architecture. They also empirically evaluate the difference in performance of the two models on community detection and counting walk problems.

To obtain the result in 1, the paper uses the recent equivalence between the computation in a depth-k GNN and the WL graph isomorphism test. The authors use this to construct two non-isomorphic graphs that the  WL test can distinguish using 2 iterations, but on which a GA-MLP will produce the same augmented embeddings. To obtain the result in 2, the authors count the number of distinct rooted trees that can be produced during the computations of a GNN and GA-MLP, and show a gap between the case of a GNN and a GA-MLP.

My main main concern with the paper is that it does not provide sufficiently rigorous findings on either the theoretical or the experimental front.   On the theoretical side, the paper establishes a somewhat expected performance gap between using a full GNN and an approximation such as a GA-MLP. Also it seems that the lower bounds for GA-MLPs only hold for the variant where the linear transformations are of the form A,A^2, and so on. Perhaps the authors can clarify this point?  Further there are no sample complexity or generalization bounds provided.   On the experimental front, it would have been much nicer to understand the tradeoffs in performance versus scalability or provide guidance on which models are more suitable for common, real-world GNN problems and applications. Counting attributed walks and community detection are not very representative problems for GNNs.

---

> ### Author Response · Authors · 2020-11-24
> **Authors' response**
>
> Thank you for the helpful comments and suggestions! Below are our responses to the two main questions:
>
> 1. _Regarding whether the theoretical lower bounds for GA-MLPs only hold for the variant consisting of powers of a generalized normalized adjacency matrix as the operator family:_
>
> This is true for Proposition 4 (that the GA-MLP induces only poly-exponential-in-depth many equivalence classes of rooted graphs) and the second half of Proposition 5 (that the GA-MLP cannot distinguish any pair of rooted subgraphs that the GNN cannot distinguish). Meanwhile, the **first half of Proposition 5** (that there exists an exponential-in-depth gap between the equivalence classes of rooted graphs induced by GNN and the GA-MLP) and **Proposition 6** (that the GA-MLP cannot exactly count attributed walks) hold for GA-MLPs with **any** family of equivariant linear operators, including any element-wise functions or matrix products of the adjacency matrix (and the compositions thereof), for example, which arguably encompasses all reasonable choices of the linear operator. In fact, in the revised edition, we prove in Appendix A that these results can be extended in arbitrary equivariant graph operators (not necessarily linear). The intuition is the following: the proofs are built upon examples of rooted graphs with a high degree of symmetry in its topology, meaning that there are many sets of nodes such that exchanging the features among the nodes in these sets will not affect the output of a GA-MLP on this rooted graph, as long as the GA-MLP uses operators that are intrinsic to the graph topology. Hence, heuristically speaking, the GA-MLP loses much more information compared to an GNN, which can be captured by the number of equivalence classes on the rooted graphs. Therefore, even for general operator choices, we can characterize the gap between the expressive powers of GA-MLPs and GNNs in these ways.
>
> 2. _Regarding what our results imply for practical applications:_
>
> Indeed, we hope to shed some light on the types of practical tasks on which GNNs have an edge over GA-MLPs. As we showed both in theory and in experiments, GA-MLPs are not expressive enough for counting attributed walks in graphs, and hence applications where this task is explicitly or implicitly involved could show a gap between GNNs and GA-MLPs. For example, in the context of knowledge graph reasoning, it has been shown in [1] that GNNs are able to encode any logical rule corresponding to a path (which is an attributed walk in our language). Based on our theory, these logical rules cannot be exactly encoded by any GA-MLP model adapted to such a task. Another example in the context of learning combinatorial algorithms is complex dynamical programming algorithms, with which GNNs are known to have good algorithmic alignment ([2]). More generally, the counting of graph substructures has been known to be relevant to graph-based tasks in molecular predictions ([3]), computational biology ([4]) as well as social network analysis ([5]), and we expect to see a gap between the performances of GNNs and GA-MLPs on many of these tasks as well. In addition, community detection is still an important problem in social network analysis, and there have been recent interests in incorporating node feature information into the task, for which GNNs can be very helpful ([6, 7]).
>
>
> References:
>
> [1] Teru, Denis, Hamilton (2020). "Inductive Relation Prediction by Subgraph Reasoning". _ICML 2020_.
>
> [2] Xu, Li, Zhang, Du, Kawarabayashi, Jegelka (2020). "What Can Neural Networks Reason About?" _ICLR 2020_.
>
> [3] Murray and Rees (2009). "The rise of fragment-based drug discovery". _Nature Chemistry_.
>
> [4] Koyutürk, Grama, Szpankowski (2004). "An efficient algorithm for detecting frequent subgraphs in biological networks." _Bioinformatics_.
>
> [5] Jiang, Coenen, Zito (2010). "Finding frequent subgraphs in longitudinal social network data using a weighted graph mining approach." _International Conference on Advanced Data Mining and Applications_.
>
> [6] Chunaev (2020). "Community detection in node-attributed social networks: a survey". _Computer Science Review_.
>
> [7] Chen, Li, Bruna (2019). "Supervised community detection with line graph neural networks". _ICLR 2019_.

---

### Official Review · AnonReviewer2 · 2020-10-29
**Expressvity of graph MLPs with operators**

**Rating:** 7
**Confidence:** 4

**Review:**

The paper compares graph neural networks (GNNs) with graph-augmented multi-layer perceptrons (GA-MLPs) where GA-MLPs are MLPs over nodes with additional node features computed over the graph. The paper contains theoretical results and experimental results for graph isomorphism testing and for node level functions. In the overflow of papers studying the expressivity of GNNs, the originality comes from the study of GA-MLPs and from the comparison for node level functions.

The paper is well written and there is a nice balance between theoretical results and empirical ones. Overall the results are not surprising but, to the best of my knowledge, they are not written elsewhere and the proofs are given. Thus, in my opinion, the paper is an original contribution to the field and can be published. I think that the paper could be improved by stating clearly (at the beginning of the paper) that the expressive power of GA-MLPs heavily rely on the choice of the operator family. And, along the paper, it should be made clear when considering GA-MLPs which class of GA-MLPs is considered (for instance, the class GA-MLPA is introduced in Section 6). Also, a synthetic comparison of the results for GA-MLPs w.r.t. the operator family could be given in the paper.

Please note that I read the proofs but I could not check them in detail.

Comments.

Introduction. In my opinion, there should be a more detailed description of graph operators used in the paper as well as the classes of GA-MLPs considered in the paper. Indeed, as such, the proposed contributions are not precise enough and can be misleading.

Section 3.3. Choices for $\Omega$ used in the paper should be presented here. Perhaps, classes of GA-MLPs should be named accordingly.

Section 4. Only consider GA-MLPs. Already Prop.1, Prop. 2 and last § show the importance of the operator family for GA-MLPs.

Section 5. Could be divided in two sections: one for GNNs and one for GA-MLPs. For GA-MLPs, again there is a clear hierarchy depending on the operator family.

Section 6.2. GA-MLP mean GA-MLPA ?

Section 6.3. The end of the section clearly shows the importance of the choice of the operator family and this should have been made clear before.


Details and Typos
* §Depth in GNNs. the their
* Not easy to find references in Appendix: ref to Appendix J is given before the proposition. Reference to Appendix E is not given.
* Section 5 §2. "analogous to the equivalence relations on $\cal G$ introduced in Section 4". I do not see such relations in Section 4.
* Appendix F is "Lemma 3"

---

> ### Author Response · Authors · 2020-11-24
> **Authors' response**
>
> Thank you for the careful comments and suggestions!
>
> Indeed, you were right in pointing out that several of our results are specific to certain choices of the operators, and we will make sure to state this clearly in the paper. In section 6.2, by “GA-MLP” indeed we meant “GA-MLP-A”. At the same time, let us highlight that two of our theoretical bounds hold for _arbitrary_ equivariant linear operators on the graph, including the first half of Proposition 5 (that there exists an exponential-in-depth gap between the equivalence classes of rooted graphs induced by GNN and the GA-MLP) and Proposition 6 (that the GA-MLP cannot exactly count attributed walks). In fact, in the revised edition, we prove in Appendix A that these results can be extended in arbitrary equivariant graph operators (not necessarily linear). Therefore, even for general choices of the operators, we can still characterize the gap between the expressive powers of GA-MLPs and GNNs in these ways.
>
> In addition, in the revised edition, we also added another GA-MLP model, GA-MLP-$\tilde{A}$, with a different choice of the operator family to the experiments of counting attributed walks and community detection for comparison. Its operator family consists of powers of the normalized adjacency matrix. The results are given in Table 2 and Table 4 in the revised version. We see that it shows no improvement in either task compared to GA-MLP-A, consistent with our theoretical results that GA-MLPs are unable to count attributed walks or learn the optimal operators (the Bethe Hessians) for community detection.
>
> Finally, thanks a lot for spotting the typos!

---

### Author Response · Authors · 2020-11-24
**A revised version is uploaded**

Dear reviewers,

We sincerely appreciate your efforts in reviewing our paper! We have updated the paper with a revised edition, with mainly the following changes:

1. We extended the definition of GA-MLPs to a general one, which allows a general (not necessarily linear) operator on the graph for the step of node feature augmentation (as suggested by Reviewer 1). Moreover, we proved that the **first half of Proposition 5** (that there exists an exponential-in-depth gap between the equivalence classes of rooted graphs induced by GNN and the GA-MLP) and **Proposition 6** (that the GA-MLP cannot exactly count attributed walks) hold for this more general type of GA-MLP as well. Due to limitation of space, these results are given in Appendix A.

2. We also added another GA-MLP model, GA-MLP-$\tilde{A}_{(1)}$, with a different choice of the operator family to the experiments of counting attributed walks and community detection for comparison. Its operator family consists of powers of the normalized adjacency matrix. The results are given in Table 2 and Table 4 in the revised version, and also listed below. We see that it shows no improvement in either task compared to GA-MLP-A, consistent with our theoretical results that GA-MLPs are unable to count attributed walks or learn the optimal operators (the Bethe Hessians) for community detection.

- Counting attributed walks (smaller loss means better performance)

GA-MLP-$A$(+) stands for GA-MLP(+) in the original submission.

|                   | Cora    | Cora    | RRG     | RRG     |
|-------------------|---------|---------|---------|---------|
|                   | Train   | Test    | Train   | Test    |
| GIN               | 3.98E-6 | 9.72E-7 | 3.39E-5 | 2.61E-4 |
| GA-MLP-$A$          | 1.23E-1 | 1.56E-1 | 1.75E-2 | 2.13E-2 |
| GA-MLP-$A$+         | 1.87E-2 | 6.44E-2 | 1.69E-2 | 2.13E-2 |
| GA-MLP-$\tilde{A}_{(1)}$  | 4.22E-1 | 5.79E-1 | 1.02E-1 | 1.58E-1 |
| GA-MLP-$\tilde{A}_{(1)}$+ | 4.00E-1 | 5.79E-1 | 1.12E-1 | 1.52E-1 |

- Community detection (higher overlap means better performance)

| Rank of Hardness | 1     | 2     | 3     | 4     | 5     |
|------------------|-------|-------|-------|-------|-------|
| GA-MLP-$H$         | 0.165 | 0.247 | 0.424 | 0.763 | 0.939 |
| GA-MLP-$A$        | 0.164 | 0.222 | 0.362 | 0.709 | 0.936 |
| GA-MLP-$\tilde{A}_{(1)}$ | 0.128 | 0.164 | 0.262 | 0.707 | 0.563 |

3. Fixed some typos.

---

### Decision · Program_Chairs · 2021-01-07
**Final Decision**

**Decision:**

Accept (Poster)

**Comment:**

This paper compares “Graph Augmented MLPs” (GA-MLP), which augment node features by a single aggregation of neighbors and then pass the resulting features through an MLP, to graph neural networks (GNNs). The paper establishes results on representational power of some GA-MLP models being less powerful than GNNs. While practitioners may not change their behavior as a result, the work appears carefully done, is novel, and reviewers are mostly in agreement that the paper is a nice read and good contribution to the field.